# Meta-Learning with Fewer Tasks through Task Interpolation

**Huaxiu Yao**[1]**, Linjun Zhang**[2]**, Chelsea Finn**[1]
[1]Stanford University, [2]Rutgers University
[1]`{huaxiu,cbfinn}@cs.stanford.edu`, [2]`linjun.zhang@rutgers.edu`

## Abstract

Meta-learning enables algorithms to quickly learn a newly encountered task with just a few labeled examples by transferring previously learned knowledge. However, the bottleneck of current meta-learning algorithms is the requirement of a large number of meta-training tasks, which may not be accessible in real-world scenarios. To address the challenge that available tasks may not densely sample the space of tasks, we propose to augment the task set through interpolation. By meta-learning with task interpolation (MLTI), our approach effectively generates additional tasks by randomly sampling a pair of tasks and interpolating the corresponding features and labels. Under both gradient-based and metric-based meta-learning settings, our theoretical analysis shows MLTI corresponds to a data-adaptive meta-regularization and further improves the generalization. Empirically, in our experiments on eight datasets from diverse domains including image recognition, pose prediction, molecule property prediction, and medical image classification, we find that the proposed general MLTI framework is compatible with representative meta-learning algorithms and consistently outperforms other state-of-the-art strategies.

## 1 Introduction

Meta-learning has powered machine learning systems to learn new tasks with only a few examples, by learning how to learn across a set of meta-training tasks. While existing algorithms are remarkably efficient at adapting to new tasks at meta-test time, the meta-training process itself is not efficient. Analogous to the training process in supervised learning, the meta-training process treats tasks as data samples and the superior performance of these meta-learning algorithms relies on having a large number of diverse meta-training tasks. However, sufficient meta-training tasks may not always be available in real-world. Take medical image classification as an example: due to concerns of privacy, it is impractical to collect large amounts of data from various diseases and construct the meta-training tasks. Under the task-insufficient scenario, the meta-learner can easily memorize these meta-training tasks, limiting its generalization ability on the meta-testing tasks. To address this limitation, we aim to develop a strategy to regularize meta-learning algorithms and improve their generalization when the meta-training tasks are limited and only sparsely cover the space of relevant tasks.

Recently, a variety of regularization methods for meta-learning have been proposed, including techniques that impose explicit regularization to the meta-learning model (Jamal and Qi, 2019; Yin et al., 2020) and methods that augment tasks by making modifications to individual training tasks through noise (Lee et al., 2020) or mixup (Ni et al., 2021; Yao et al., 2021). However, these methods are largely designed to either tackle only the memorization problem (Yin et al., 2020) or to improve performance of meta-learning (Yao et al., 2021) when plenty of meta-training tasks are provided. Instead, we aim to target the task distribution directly, leading to an approach that is particularly well-suited to settings with limited meta-training tasks.

Concretely, as illustrated in Figure 1, we aim to densify the task distribution by providing interpolated tasks across meta-training tasks, resulting in a new task interpolation algorithm named **MLTI** (**M**eta-**L**earning with **T**ask **I**nterpolation). The key idea behind MLTI is to generate new tasks by interpolating between pairs of randomly sampled meta-training tasks. This interpolation can be instantiated in a variety of ways, and we present two variants that we find to be particularly effective. The first label-

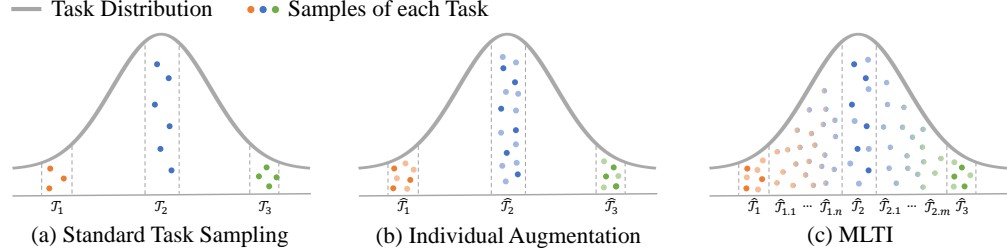

Figure 1: Motivations behind MLTI. (a) three tasks are sampled from the task distribution; (b) individual augmentation methods (e.g., (Ni et al., 2021; Yao et al., 2021) augment each task within its own distribution); (c) MLTI densifies the task-level distribution by performing cross-task interpolation.

sharing (LS) scenario includes tasks that share the same set of classes (e.g., RainbowMNIST (Finn et al., 2019)). For each LS task pair randomly drawn from the meta-training tasks, MLTI linearly interpolates their features and accordingly applies the same interpolation strategy on the corresponding labels. The second non-label-sharing (NLS) scenario includes classification tasks with different sets of classes (e.g., miniImagenet). For each additional NLS task, we first randomly select two original meta-training tasks and then generate new classes by linearly interpolating the features of the sampled classes, which draw one class in each original task without replacement. Since MLTI is essentially changing only the tasks, it can be readily used with any meta-learning approach and can be combined with prior regularization techniques that target the model.

In summary, our primary contributions are: (1) We propose a new task augmentation method (MLTI) that densifies the task distribution by introducing additional tasks; (2) Theoretically, we prove that MLTI regularizes meta-learning algorithms and improves the generalization ability. (3) Empirically, in eight real-world datasets from various domains, MLTI consistently outperforms six prior meta-learning regularization methods and is compatible with six representative meta-learning algorithms.

## 2 PRELIMINARIES

**Problem statement.** In meta-learning, we assume each task $\mathcal{T}_i$ is $i.i.d.$ sampled from a task distribution $p(\mathcal{T})$ associated with a dataset $\mathcal{D}_i$, from which we $i.i.d.$ sample a support set $\mathcal{D}_i^s = (\mathbf{X}_i^s, \mathbf{Y}_i^s) = \{(\mathbf{x}_{i,k}^s, \mathbf{y}_{i,k}^s)\}_{k=1}^{N_s}$ and a query set $\mathcal{D}_i^q = (\mathbf{X}_i^q, \mathbf{Y}_i^q) = \{(\mathbf{x}_{i,k}^q, \mathbf{y}_{i,k}^q)\}_{k=1}^{N_q}$. Given a predictive model $f$ (a.k.a., the base model) with parameter $\theta$, meta-learning algorithms first train the base model on meta-training tasks. Then, during the meta-testing stage, the well-trained base model $f$ is applied to the new task $\mathcal{T}_t$ with the help of its support set $\mathcal{D}_t^s$ and finally evaluate the performance on the query set $\mathcal{D}_t^q$. In the rest of this section, we will introduce both gradient-based and metric-based meta-learning algorithms. For simplicity, we omit the subscript of the meta-training task index $i$ in the rest of this section.

**Gradient-based meta-learning.** In gradient-based meta-learning, we use model-agnostic meta-learning (MAML) (Finn and Levine, 2018) as an example and denote the corresponding base model as $f^{MAML}$. Here, the goal of MAML is to learn initial parameters $\theta^*$ such that one or a few gradient steps on $\mathcal{D}^s$ leads to a model that performs well on task $\mathcal{T}$. During the meta-training stage, the performance of the adapted model $f_\phi$ is evaluated on the corresponding query set $\mathcal{D}^q$ and is used to optimize the model parameter $\theta$. Formally, the bi-level optimization process with expected risk is formulated as:

$$\theta^* \leftarrow \arg\min_\theta \mathbb{E}_{\mathcal{T} \sim p(\mathcal{T})} \left[ \mathcal{L}(f_\phi^{MAML}; \mathcal{D}^q) \right], \text{ where } \phi = \theta - \eta \nabla_\theta \mathcal{L}(f_\theta^{MAML}; \mathcal{D}^s), \qquad (1)$$

where $\eta$ denotes the inner-loop learning rate and $\mathcal{L}$ is defined as the loss, which is formulated as cross-entropy loss (i.e., $\mathbb{E}_{\mathcal{T} \sim p(\mathcal{T})}[-\sum_k \log p(\mathbf{y}_k^q | \mathbf{x}_k^q, f_\phi)]$) and mean square error (i.e., $\mathbb{E}_{\mathcal{T} \sim p(\mathcal{T})}[\sum_k \|f_\phi(\mathbf{x}_k^q) - \mathbf{y}_k^q\|]$) for classification and regression problems, respectively. During the meta-testing stage, for task $\mathcal{T}_t$, the adapted parameter $\phi_t$ is achieved by fine-tuning $\theta_t$ on the support set $\mathcal{D}_t^s$.

**Metric-based meta-learning.** The aim of metric-based meta-learning is to perform a non-parametric learner on the top of meta-learned embedding space. Taking prototypical network (ProtoNet) with base model $f^{PN}$ as an example (Snell et al., 2017), for each task $\mathcal{T}$, we first compute class prototype representation $\{\mathbf{c}_r\}_{r=1}^R$ as the representation vector of the support samples belonging to class $k$ as $\mathbf{c}_r = \frac{1}{N_r} \sum_{(\mathbf{x}_{k;r}^s, \mathbf{y}_{k;r}^s) \in \mathcal{D}_r^s} f_\theta^{PN}(\mathbf{x}_{k;r}^s)$, where $\mathcal{D}_r^s$ represents the subset of support samples labeled as

class $r$ and the number of this subset is $N_r$. Then, given a query data sample $\mathbf{x}_k^q$ in the query set, the probability of assigning it to the $r$-th class is measured by the distance $d$ between its representation $f_\theta^{PN}(\mathbf{x}_k^q)$ and prototype representation $\mathbf{c}_r$, and the cross-entropy loss of ProtoNet is formulated as:

$$\mathcal{L} = \mathbb{E}_{\mathcal{T} \sim p(\mathcal{T})} \left[ -\sum_{k,r} \log p(\mathbf{y}_k^q = r | \mathbf{x}_k^q) \right] = \mathbb{E}_{\mathcal{T} \sim p(\mathcal{T})} \left[ -\sum_{k,r} \log \frac{\exp(-d(f_\theta^{PN}(\mathbf{x}_k^q), \mathbf{c}_r))}{\sum_{r'} \exp(-d(f_\theta^{PN}(\mathbf{x}_k^q), \mathbf{c}_{r'}))} \right]. \quad (2)$$

At the meta-testing stage, the predicted label of each query samples is assigned to the class with maximal probability (i.e., $\hat{\mathbf{y}}_k^q = \arg\max_r p(\mathbf{y}_k^q = r | \mathbf{x}_k^q)$).

The estimation of the expected loss in Eqn. (1) or (2) is challenging since the distribution $p(\mathcal{T})$ is unknown in practical situations. A common way of estimation is to approximate the expected risk in Eqn. (1) by a set of meta-training tasks $\{\mathcal{T}_i\}_{i=1}^{|I|}$ (use MAML as an example):

$$\theta^* \leftarrow \frac{1}{|I|} \arg\min_\theta \sum_{i=1}^{|I|} \mathcal{L}(f_{\phi_i}^{MAML}; \mathcal{D}_i^q), \quad \text{where } \phi_i = \theta - \alpha \nabla_\theta \mathcal{L}(f_\theta^{MAML}; \mathcal{D}_i^s). \quad (3)$$

However, this approximation method still faces the challenge: optimizing Eqn. (3), as suggested in (Rajendran et al., 2020; Yin et al., 2020), can result in memorization of the meta-training tasks, thus limiting the generalization of the meta-learning model to new tasks, especially in domains with limited meta-training tasks.

## 3 META-LEARNING WITH TASK INTERPOLATION

To address the memorization issue described in the last section, we aim to develop a framework that allows meta-learning methods to generalize well to new few-shot learning tasks, even when the provided meta-training tasks are only sparsely sampled from the task distribution. To accomplish this, we introduce meta-learning with task interpolation (MLTI). The key idea behind MLTI is to *densify* the task distribution by generating new tasks that interpolate between provided meta-training tasks. This approach requires no additional task data or supervision, and can be combined with any base meta-learning algorithm, including MAML and ProtoNet.

Before detailing the proposed strategy, we first discuss two scenarios of meta-training task distributions, *label-sharing* and *non-label-sharing* tasks, which have distinct implications for task interpolation. Formally, we define these two scenarios as:

**Definition 1 (label-sharing tasks)** *If the labels of all tasks share the same label space, we refer it as the label-sharing (LS) scenario. Take Pascal3D pose prediction (Yin et al., 2020) as an example, each task is to predict the current orientation of the object relative to its canonical orientation, and the range of canonical orientation is shared across all tasks.*

**Definition 2 (non-label-sharing tasks)** *The non-label-sharing (NLS) scenario assumes that different semantic meanings of labels across tasks. For example, the* piano *class in the miniImagenet dataset may correspond to a class label of* 0 *for one task and* 1 *for another task.*

**MLTI for label-sharing tasks**. First, we will discuss MLTI under the label-sharing scenario, where it applies the same interpolation strategy on both features/hidden representations and label spaces. Concretely, let's say that a model $f$ consists of $L$ layers and the hidden representation of samples $\mathbf{X}$ at the $l$-th layer is denoted as $\mathbf{H}^l = f_{\theta^l}(\mathbf{X})$ ($0 \leq l \leq L^s$), where $\mathbf{H}^0 = \mathbf{X}$ and $L^s$ represents the number of layers shared across all tasks. In gradient-based methods, as suggested in (Yin et al., 2020), only part of the layers are shared (i.e., $L^s < L$). In metric-based methods, all layers are shared (i.e., $L^s = L$). Given a pair of tasks with their sampled support and query sets (i.e., $\mathcal{T}_i = \{\mathcal{D}_i^s, \mathcal{D}_i^q\}$ and $\mathcal{T}_j = \{\mathcal{D}_j^s, \mathcal{D}_j^q\}$) under the same label space, MLTI first randomly selects one layer $l$ and then applies the task interpolation separately on the hidden representations ($\mathbf{H}_i^{s(q),l}$, $\mathbf{H}_j^{s(q),l}$) and corresponding labels ($\mathbf{Y}_i^{s(q)}$, $\mathbf{Y}_j^{s(q)}$) of the support (query) sets as:

$$\tilde{\mathbf{H}}_{cr}^{s,l} = \lambda \mathbf{H}_i^{s,l} + (1-\lambda)\mathbf{H}_j^{s,l}, \quad \tilde{\mathbf{Y}}_{cr}^{s,l} = \lambda \mathbf{Y}_i^s + (1-\lambda)\mathbf{Y}_j^s,$$
$$\tilde{\mathbf{H}}_{cr}^{q,l} = \lambda \mathbf{H}_i^{q,l} + (1-\lambda)\mathbf{H}_j^{q,l}, \quad \tilde{\mathbf{Y}}_{cr}^{q,l} = \lambda \mathbf{Y}_i^q + (1-\lambda)\mathbf{Y}_j^q, \quad (4)$$

where $\lambda \in [0,1]$ is sampled from a Beta distribution $\text{Beta}(\alpha, \beta)$ and the subscript "cr" represents "cross". Notice that both the support and query sets will be replaced by the interpolated ones in

MLTI, while only the query set is replaced in approaches like Yao et al. (2021). Besides, Manifold Mixup (Verma et al., 2019) in Eqn. (4) can be replaced by different task interpolation methods (e.g., Mixup (Zhang et al., 2018), CutMix (Yun et al., 2019)).

**MLTI for non-label-sharing tasks.** Under non-label-sharing scenarios, tasks have different label spaces, making it infeasible to directly interpolate the labels. Instead, we generate the new task by performing the feature-level interpolation and re-assign a new label to the interpolated class. Specifically, given samples from class $r$ in task $\mathcal{T}_i$ and class $r'$ in task $\mathcal{T}_j$, we denote the interpolated features as $\text{Intrpl}(r, r')$, which are formally defined as:

$$\tilde{\mathbf{H}}^{s,l}_{cr;r} = \lambda \mathbf{H}^{s,l}_{i;r} + (1 - \lambda)\mathbf{H}^{s,l}_{j;r'}, \quad \tilde{\mathbf{H}}^{q,l}_{cr;r} = \lambda \mathbf{H}^{q,l}_{i;r} + (1 - \lambda)\mathbf{H}^{q,l}_{j;r'}. \tag{5}$$

The interpolated samples are regarded as a new class in the interpolated task. After randomly selecting $N$ class pairs, we can construct an $N$-way interpolated task. Take a 3-way classification as an example, assume task $\mathcal{T}_i$ has classes $(i_1, i_2, i_3)$ and task $\mathcal{T}_j$ has classes $(j_1, j_2, j_3)$. One potential interpolated task could be a 3-way task with classes $(e_1, e_2, e_3)$, where the labels are associated with interpolated features $(\text{Intrpl}(i_1, j_2), \text{Intrpl}(i_2, j_3), \text{Intrpl}(i_3, j_1))$. Note that, for ProtoNet and its variants, we apply the interpolation strategies of Eqn. (5) on both LS and NLS scenarios since it is intractable to calculate prototypes with mixed labels.

Finally, we note that MLTI supports both inter-task and intra-task interpolation, as we allow the case when $i = j$. As we will find in Sec. 6, intra-task interpolation can be complementary to cross-task interpolation and further improve the generalization. Under this case, the intra-task interpolation can also be replaced by any existing intra-task augmentation strategies (e.g., MetaMix (Yao et al., 2021)).

After generating the interpolated support set $\mathcal{D}^s_{i,cr} = (\tilde{\mathbf{H}}^{s,l}_{i,cr}, \tilde{\mathbf{Y}}^s_{i,cr})$ and query set $\mathcal{D}^q_{i,cr} = (\tilde{\mathbf{H}}^{q,l}_{i,cr}, \tilde{\mathbf{Y}}^q_{i,cr})$, we replace the original support and query sets with the interpolated ones. With MAML as an example, we reformulate the optimization process in Eqn. (3) as:

$$\theta^* \leftarrow \frac{1}{|I|} \arg\min_\theta \sum_{i=1}^{|I|} \mathcal{L}(f^{MAML}_{\phi^{L-l}_{i,cr}}; \mathcal{D}^q_{i,cr}), \quad \text{where } \phi^{L-l}_{i,cr} = \theta^{L-l} - \alpha\nabla_{\theta^{L-l}}\mathcal{L}(f^{MAML}_{\theta^{L-l}}; \mathcal{D}^s_{i,cr}), \tag{6}$$

where the superscript $L - l$ represents the rest of layers after the selected layer $l$. Detailed pseudocode of MAML and ProtoNet is shown in Alg. 1 and Alg. 2 in Appendix A, respectively.

## 4 THEORETICAL ANALYSIS

We now theoretically investigate how MLTI improves the generalization performance with both gradient-based and the metric-based meta-learning methods. Specifically, we theoretically prove that MLTI essentially induces a data-dependent regularizer on both categories of meta-learning methods and controls the Rademacher complexity, leading to greater generalization. Here, we only discuss the non-label-sharing (NLS) scenario (see detailed proof in Appendix B.1) and leave the analysis of the label-sharing scenario in Appendix B.2.

### 4.1 GRADIENT-BASED META-LEARNING WITH MLTI

In gradient-based meta-learning, we analyze the generalization ability by considering the two-layer neural network with binary classification. For the simplicity of presentation, we assume the sample size of different task are the same and equal to $N$. Suppose there are $|I|$ tasks. For each task $\mathcal{T}_i$, we consider the logistic loss $\ell(f^{MAML}(\mathbf{x}), \mathbf{y}) = \log(1 + \exp(f^{MAML}(\mathbf{x}))) - yf^{MAML}(\mathbf{x})$ with $f^{MAML}$ modeled by $f^{MAML}_{\phi_i}(\mathbf{x}_{i,k}) = \phi_i^\top \sigma(\mathbf{W}\mathbf{x}_{i,k}) := \phi_i^\top \mathbf{h}^1_{i,k}$, where $\mathbf{h}^1_{i,k}$ represents the hidden representation on the first layer of sample $\mathbf{x}_{i,k}$. Under the NLS setting, the interpolated task is constructed by Eqn. (5). We assume the interpolation performs on the hidden layer (following Eqn. (5) with $l = 1$) and denote the interpolated query set as $\mathcal{D}^q_{i,cr} = (\tilde{\mathbf{H}}^{q,1}_{i,cr}, \tilde{\mathbf{Y}}^q_{i,cr})$. For simplicity, in this subsection, we omit the superscript $q$ and define the empirical training loss as $\mathcal{L}_t(\{\mathcal{D}_{i,cr}\}_{i=1}^{|I|}) = |I|^{-1}\sum_{i=1}^{|I|}\mathcal{L}(\mathcal{D}_{i,cr}) = (N|I|)^{-1}\sum_{i=1}^{|I|}\sum_{k=1}^{N}\mathcal{L}(f_{\phi_i}(\mathbf{x}_{i,k,cr}), \mathbf{y}_{i,k,cr})$. We first present a lemma showing that the loss $\mathcal{L}_t(\{\mathcal{D}_{i,cr}\}_{i=1}^{|I|})$ induced by MLTI has a regularization effect.

**Lemma 1.** *Consider the MLTI with $\lambda \sim \text{Beta}(\alpha, \beta)$. Let $\psi(u) = e^u/(1 + e^u)^2$ and $N_{i,r}$ denotes the number of samples from the class $r$ in task $\mathcal{T}_i$. There exists a constant $c > 0$, such that the*

*second-order approximation of $\mathcal{L}_t(\{\mathcal{D}_{i,cr}\}_{i=1}^{|I|})$ is given by*

$$\mathcal{L}_t(\bar{\lambda} \cdot \{\mathcal{D}_i\}_{i=1}^{|I|}) + c\frac{1}{N|I|}\sum_{i=1}^{|I|}\sum_{k=1}^{N}\psi(\mathbf{h}_{i,k}^{1\top}\phi_i)\cdot\phi_i^\top(\frac{1}{|I|}\sum_{i=1}^{|I|}\frac{1}{2}\sum_{r=1}^{2}\frac{1}{N_{i,r}}\sum_{i=1}^{|I|}\sum_{k=1}^{N_{i,r}}\mathbf{h}_{i,k;r}^1\mathbf{h}_{i,k;r}^{1\top})\phi_i, \quad (7)$$

*where $\bar{\lambda} = \mathbb{E}_{\mathcal{D}_\lambda}[\lambda]$, with $\mathcal{D}_\lambda \sim \frac{\alpha}{\alpha+\beta}\text{Beta}(\alpha+1,\beta) + \frac{\beta}{\alpha+\beta}\text{Beta}(\beta+1,\alpha)$.*

This lemma suggests that MLTI induces an (implicit) regularization term on $\phi_i$'s through task interpolation and therefore will lead to a better generalization bound, as we will show in Section 6.2 with extensive numerical experiments. To study the improved generalization more explicitly, we consider the population version of the regularization term in Eqn. (7) by considering the function class $\mathcal{F}_\gamma = \{\mathbf{H}^{1\top}\phi : \mathbb{E}[\psi(\mathbf{H}^{1\top}\phi)]\phi^\top\Sigma\phi \leq \gamma\}$, where $\Sigma = \mathbb{E}_{\mathcal{T}\sim p(\mathcal{T})}\mathbb{E}_\mathcal{T}[\mathbf{H}^1\mathbf{H}^{1\top}]$. We also define $\mu_\mathcal{T} = \mathbb{E}_\mathcal{T}[\mathbf{H}^1]$ and assume the following condition of the individual task distribution $\mathcal{T}$ as: for all $\mathcal{T} \sim p(\mathcal{T})$, $\mathcal{T}$ satisfies

$$rank(\Sigma) \leq R, \ \|\Sigma^{\dagger/2}\mu_\mathcal{T}\| \leq U, \quad (8)$$

where $\Sigma^\dagger$ denotes the generalized inverse of $\Sigma$. Further, we assume that the distribution of $\mathbf{H}^1$ is $\rho$-*retentive* for some $\rho \in (0, 1/2]$, that is, if for any non-zero vector $v \in \mathbb{R}^d$, $[\mathbb{E}[\psi(v^\top\mathbf{H}^1)]]^2 \geq \rho \cdot \min\{1, \mathbb{E}(v^\top\mathbf{H}^1)^2\}$. Such an assumption has been similarly assumed in (Arora et al., 2020; Zhang et al., 2021) and is satisfied when the weights has bounded $\ell_2$ norm.

We also regard $\mathcal{L}_t(\{\mathcal{D}_i\}_{i=1}^{|I|})$ of tasks $\{\mathcal{T}_i\}_{i=1}^{|I|}$ as the empirical (training) loss $\mathcal{R}(\{\mathcal{D}_i\}_{i=1}^{|I|})$ and its corresponding population loss (on the test data) is defined as $\mathcal{R} = \mathbb{E}_{\mathcal{T}_i\sim p(\mathcal{T})}\mathbb{E}_{(\mathbf{X}_i,\mathbf{Y}_i)\sim\mathcal{T}_i}[\mathcal{L}(f_{\phi_i}(\mathbf{X}_i),\mathbf{Y}_i)]$. We then have the following theorem showing the improved generalization gap brought by MLTI.

**Theorem 1.** *Suppose $\mathbf{X}_i$'s, $\mathbf{Y}'_i$s and $\phi$ are bounded in spectral norm and assumption (8) holds. There exist constants $A_1, A_2, A_3 > 0$, such that for all $f_\mathcal{T} \in \mathcal{F}_\gamma, \delta \in (0,1)$, with probability at least $1 - \delta$ (over randomness of training sample), we have the following generalization bound*

$$|\mathcal{R}(\{\mathcal{D}_i\}_{i=1}^{|I|}) - \mathcal{R}| \leq A_1 \max\{(\frac{\gamma}{\rho})^{1/4}, (\frac{\gamma}{\rho})^{1/2}\}(\sqrt{\frac{R+U}{N}} + \sqrt{\frac{R+U}{|I|}}) + A_2\sqrt{\frac{\log(|I|/\delta)}{N}} + A_3\sqrt{\frac{\log(1/\delta)}{|I|}}.$$

Based on Lemma 1 and Theorem 1, MLTI regularizes on $\phi^\top\Sigma\phi$ (implying a small $\gamma$) and therefore achieves a tighter generalization bound than the vanilla gradient-based method (where $\gamma$ is unconstrained). Compared with the individual task augmentation (see Figure 1(b)), the regularization effect in Eqn. (7) induced by MLTI is larger (i.e., smaller $\gamma$) since the total variance is generally larger than the within-group variance (see more details in the Appendix B.3). Therefore, MLTI reduces the generalization error, which we also empirically validate in the experiments.

### 4.2 METRIC-BASED META-LEARNING WITH MLTI

In the metric-based meta-learning, we consider the ProtoNet with linear representation in the binary classification, which has been commonly considered in other theoretical analysis of meta-learning, see, e.g., (Du et al., 2020; Tripuraneni et al., 2020). Specifically, we assume $f_\theta^{PN}(\mathbf{x}) = \theta^\top\mathbf{x}$ and $d(\cdot,\cdot)$ represents the squared Euclidean distance, then the loss of ProtoNet can be simplified as

$$\arg\min_\theta \sum_{i=1}^{|I|}\sum_{k=1}^{N}\log p(\mathbf{y}_{i,k}=r|\mathbf{x}_{i,k}) = \arg\min_\theta \sum_{i=1}^{|I|}\sum_{k=1}^{N}\frac{1}{1+\exp(\langle(\mathbf{x}_{i,k}-(\mathbf{c}_1+\mathbf{c}_2)/2,\theta\rangle)}, \quad (9)$$

where $\mathbf{c}_1$ and $\mathbf{c}_2$ are defined as the prototypes of class 1 and 2, respectively. Under this setting, the interpolation performs on the feature (i.e., $l = 0$ in Eqn. (5)).

We now present the following lemma showing that MLTI induces a regularization on the parameter $\theta$.

**Lemma 2.** *Considering the interpolated tasks $\{\mathcal{D}_{i,cr}\}_{i=1}^{|I|}$ with $\lambda \sim \text{Beta}(\alpha,\beta)$, we define $\mathcal{L}_t(\{\mathcal{D}_i\}_{i=1}^{|I|}) = (N|I|)^{-1}\sum_{i,k}(1+\exp(\langle(\mathbf{x}_{i,k}-(\mathbf{c}_1+\mathbf{c}_2)/2,\theta\rangle))^{-1}$ and $\mathcal{L}_t(\{\mathcal{D}_{i,cr}\}_{i=1}^{|I|}) = (N|I|)^{-1}\sum_{i,k}(1+\exp(\langle(\mathbf{x}_{i,k,cr}-(\mathbf{c}_{1,cr}+\mathbf{c}_{2,cr})/2,\theta\rangle))^{-1}$. Recall $\psi(u) = e^u/(1+e^u)^2$. The second-order approximation of $\mathcal{L}_t(\{\mathcal{D}_{i,cr}\}_{i=1}^{|I|})$ is given by, for some constant $c > 0$,*

$$\mathcal{L}_t(\bar{\lambda}\{\mathcal{D}_i\}_{i=1}^{|I|}) + c\cdot\frac{1}{N|I|}\sum_{i\in I,k\in[N]}\psi(\langle\mathbf{x}_{i,k}-(\mathbf{c}_1+\mathbf{c}_2)/2,\theta\rangle)\cdot\theta^\top(\frac{1}{|I|}\sum_{i=1}^{|I|}\frac{1}{2}\sum_{r=1}^{2}\frac{1}{N_r}\sum_{i=1}^{|I|}\sum_{k=1}^{N_r}\mathbf{x}_{i,k;r}\mathbf{x}_{i,k;r}^\top)\theta.$$

$$(10)$$

Similar to the last section, we assume that the distribution of $\mathbf{x}$ is $\rho$-*retentive* for some $\rho \in (0, 1/2]$, and investigate the following function class: let $\Sigma_X = \mathbb{E}[\mathbf{x}\mathbf{x}^\top]$,

$$\mathcal{W}_\gamma := \{\mathbf{x} \to \theta^\top \mathbf{x}, \text{ such that } \theta \text{ satisfying } \mathbb{E}_\mathbf{x}\left[\psi(\langle \mathbf{x} - (\mathbf{c}_1 + \mathbf{c}_2)/2, \theta \rangle)\right] \cdot \theta^\top \Sigma_X \theta \le \gamma \}. \tag{11}$$

We then have the following theorem on the explicit generalization bound of ProtoNet.

**Theorem 2.** *Suppose $\mathbf{X}_i$'s, $\mathbf{Y}_i$'s and $\theta$ are both bounded in spectral norm, and the distribution of $\mathbf{x}$ is $\rho$-retentive and mean zero. Let $r_\Sigma = rank(\Sigma_X)$, then there exist constants $B_1, B_2, B_3 > 0$, for any $f \in \mathcal{W}_\gamma, \delta \in (0, 1)$, with probability at least $1 - \delta$ (over the training sample), such that*

$$|\mathcal{R}(\{\mathcal{D}_i\}_{i=1}^{|I|}) - \mathcal{R}| \le 2B_1 \cdot \max\{(\frac{\gamma}{\rho})^{1/4}, (\frac{\gamma}{\rho})^{1/2}\} \cdot \left( \sqrt{\frac{r_\Sigma}{|I|}} + \sqrt{\frac{r_\Sigma}{N}} \right) + B_2 \sqrt{\frac{\log(1/\delta)}{2|I|}} + B_3 \sqrt{\frac{\log(|I|/\delta)}{N}}.$$

By Theorem 2, adding MLTI into ProtoNet would induce a small value of $\gamma$ and thus improve the generalization compared to the vanilla ProtoNet. Similarly, MLTI achieves tighter generalization bound than the individual task augmentation with a larger regularization term (i.e., smaller $\gamma$).

## 5 RELATED WORK

The goal of meta-learning is to enable few-shot generalization of machine learning algorithms by transferring the knowledge acquired from related tasks. One approach is gradient-based meta-learning (Finn and Levine, 2018; Finn et al., 2017; 2018; Grant et al., 2018; Flennerhag et al., 2020; Lee and Choi, 2018; Li et al., 2017; Oh et al., 2021; Nichol and Schulman, 2018; Rajeswaran et al., 2019; Rusu et al., 2018), where the meta-knowledge is formulated to be optimization-related parameters (e.g., model initial parameters, learning rate, pre-conditioning matrix). During the meta-training stage, the model is first adapted to each task via a truncated optimization and then the optimization-related parameters are optimized by maximizing the generalization performance of the model. Another line of research is metric-based meta-learning (Cao et al., 2021; Garcia and Bruna, 2018; Liu et al., 2019; Mishra et al., 2018; Snell et al., 2017; Vinyals et al., 2016; Sung et al., 2018; Yoon et al., 2019), which meta-learns an embedding space and uses a non-parametric learner to classify samples. Unlike prior works that propose new meta-learning algorithms, this work aims to improve the task-level generalization of these algorithms and reduce the negative effect of memorization, especially when the number of meta-training tasks is limited.

To mitigate the influence of memorization and improve the generalization, one line of research focuses on directly imposing regularization on meta-learning algorithms (Guiroy et al., 2019; Jamal and Qi, 2019; Tseng et al., 2020; Yin et al., 2020). Another line of research reduces the number of adapted parameters for gradient-based meta-learning (Raghu et al., 2020; Zintgraf et al., 2019). Instead of imposing regularization strategies (i.e., objectives, dropout, less adapted parameters), our approach focuses on augmenting the set of tasks for meta-training. Prior works have proposed domain-specific techniques to generate more data by augmenting images (Chen et al., 2019) or by reconstructing tasks with latent reasoning categories for NLP-related tasks (Murty et al., 2021). Recent domain-agnostic techniques have augmented tasks by imposing label noise (Rajendran et al., 2020) or applying Mixup (Zhang et al., 2018) and its variants (e.g., Manifold Mixup (Verma et al., 2019)) to each task (Ni et al., 2021; Yao et al., 2021). Unlike these domain-agnostic augmentation strategies that applying data augmentation on each task individually (Figure 1(b)), we directly densify the task distribution by generating additional tasks from pairs of existing tasks (Figure 1(c)). More discussions with individual task augmentation are provided in Appendix C. Empirically, we find that MLTI outperforms all of these above strategies in Section 6.

## 6 EXPERIMENTS

In this section, we conduct experiments to test and understand the effectiveness of MLTI. Specifically, we aim to answer the following research questions under both label-sharing and non-label-sharing settings: **Q1**: Compared with prior methods for regularizing meta-learning, how does the MLTI perform? **Q2**: Is MLTI compatible with different backbone meta-learning algorithms and does it improve their performance? **Q3**: How does MLTI perform compared with only applying intra- or cross-task interpolation? **Q4**: How does the number of tasks affect the performance of MLTI?

Table 1: Overall performance (averaged accuracy/MSE (Pose) $\pm$ 95% confidence interval) under label-sharing scenario. MLTI consistently improves the performance under the label-sharing scenario.

| Backbone | Strategies | Pose (15-shot) | RMNIST (1-shot) | NCI (5-shot) | Metabolism (5-shot) |
|---|---|---|---|---|---|
| MAML | Vanilla | $2.383 \pm 0.087$ | $57.34 \pm 1.25\%$ | $77.09 \pm 0.85\%$ | $57.22 \pm 1.01\%$ |
| | Meta-Reg | $2.358 \pm 0.089$ | $58.10 \pm 1.15\%$ | $77.34 \pm 0.87\%$ | $58.00 \pm 0.96\%$ |
| | TAML | $2.208 \pm 0.091$ | $56.21 \pm 1.46\%$ | $76.50 \pm 0.87\%$ | $57.87 \pm 1.05\%$ |
| | Meta-Dropout | $2.501 \pm 0.090$ | $56.19 \pm 1.39\%$ | $77.21 \pm 0.82\%$ | $57.53 \pm 1.02\%$ |
| | MetaAug | $2.296 \pm 0.080$ | $55.58 \pm 0.97\%$ | $76.31 \pm 0.98\%$ | $56.65 \pm 1.00\%$ |
| | MetaMix | $2.064 \pm 0.075$ | $64.60 \pm 1.14\%$ | $76.88 \pm 0.73\%$ | $58.61 \pm 1.03\%$ |
| | Meta-Maxup | $2.107 \pm 0.077$ | $62.13 \pm 1.08\%$ | $77.90 \pm 0.79\%$ | $58.43 \pm 0.99\%$ |
| | **MLTI (ours)** | $\mathbf{1.976 \pm 0.073}$ | $\mathbf{65.92 \pm 1.17\%}$ | $\mathbf{79.14 \pm 0.73\%}$ | $\mathbf{60.28 \pm 1.00\%}$ |
| ProtoNet | MetaAug | n/a | $65.41 \pm 1.10\%$ | $74.84 \pm 0.87\%$ | $61.06 \pm 0.94\%$ |
| | MetaMix | n/a | $67.80 \pm 0.97\%$ | $75.84 \pm 0.85\%$ | $62.04 \pm 0.93\%$ |
| | Meta-Maxup | n/a | $66.18 \pm 1.08\%$ | $75.65 \pm 0.84\%$ | $61.36 \pm 0.91\%$ |
| | **MLTI (ours)** | n/a | $\mathbf{70.14 \pm 0.92\%}$ | $\mathbf{76.90 \pm 0.81\%}$ | $\mathbf{63.47 \pm 0.96\%}$ |

Table 2: Ablation Studies under label-sharing scenario. The results are reported by the averaged accuracy/MSE $\pm$ 95% confidence interval.

| Backbone | Strategies | Pose (15-shot) | RMNIST (1-shot) | NCI (5-shot) | Metabolism (5-shot) |
|---|---|---|---|---|---|
| MAML | Vanilla | $2.383 \pm 0.087$ | $57.34 \pm 1.25\%$ | $77.09 \pm 0.85\%$ | $57.22 \pm 1.01\%$ |
| | Intra-Intrpl | $2.072 \pm 0.077$ | $62.57 \pm 1.70\%$ | $78.23 \pm 0.78\%$ | $58.70 \pm 0.97\%$ |
| | Cross-Intrpl | $2.017 \pm 0.072$ | $65.34 \pm 1.78\%$ | $78.64 \pm 0.80\%$ | $59.60 \pm 1.00\%$ |
| | **MLTI** | $\mathbf{1.976 \pm 0.073}$ | $\mathbf{65.92 \pm 1.17\%}$ | $\mathbf{79.14 \pm 0.73\%}$ | $\mathbf{60.28 \pm 1.00\%}$ |
| ProtoNet | Vanilla | n/a | $65.41 \pm 1.10\%$ | $74.84 \pm 0.87\%$ | $61.06 \pm 0.94\%$ |
| | Intra-Intrpl | n/a | $67.32 \pm 0.94\%$ | $75.26 \pm 0.87\%$ | $61.66 \pm 0.88\%$ |
| | Cross-Intrpl | n/a | $69.97 \pm 0.85\%$ | $76.32 \pm 0.85\%$ | $62.48 \pm 0.91\%$ |
| | **MLTI** | n/a | $\mathbf{70.14 \pm 0.92\%}$ | $\mathbf{76.90 \pm 0.81\%}$ | $\mathbf{63.47 \pm 0.96\%}$ |

We compare MLTI with the following two representative domain-agnostic strategies: (1) directly imposing regularization into the meta-learning framework, including Meta-Reg (Yin et al., 2020), TAML (Jamal and Qi, 2019), and Meta-dropout (Lee et al., 2020); and (2) individual task augmentation methods, including Meta-Augmentation (Rajendran et al., 2020), MetaMix (Yao et al., 2021), and Meta-Maxup (Ni et al., 2021). We select MAML and ProtoNet as backbone methods and apply the corresponding meta-learning strategies to them according to their applicable scopes. Note that we also extend MetaMix and Meta-Maxup to ProtoNet, even though the methods only focus on gradient-based meta-learning in the original papers. To further test the compatibility of MLTI, we additionally apply MLTI to other meta-learning backbone algorithms, including MetaSGD (Li et al., 2017), ANIL (Raghu et al., 2020), Meta-Curvature (MC) (Park and Oliva, 2019), and MatchingNet (Vinyals et al., 2016). To provide a fair comparison, all methods use the same architecture of the base model as MLTI and all interpolation-based methods use the same interpolation strategies (see Appendix D.1 and E.1 for details).

## 6.1 LABEL-SHARING SCENARIO

**Datasets and experimental setup.** Under the label-sharing scenario, we perform experiments on four datasets to evaluate the performance of MLTI: (1) PASCAL3D Pose regression (Pose) (Yin et al., 2020): it aims to predict the object pose of a grey-scale image relative to the canonical orientation. Following Yin et al. (2020), we select 50 objects for meta-training and 15 objects for meta-testing; (2) RainbowMNIST (RMNIST) (Finn et al., 2019): it is a 10-way classification dataset wherein each task is constructed by applying a combination of image transformation operators on the original MNIST dataset (e.g., scaling, coloring, rotation). We here use 14 and 10 combinations for meta-training and meta-testing, respectively. (3)&(4) NCI (NCI, 2018) and TDC Metabolism (Metabolism) (Huang et al., 2021): both are 2-way chemical classification datasets, which aim to predict the property of a set of chemical compounds. We use six data sources for meta-training, and the remaining three sources for meta-testing. The number of shots for the above four datasets are set as 15, 1, 5, and 5,

Table 3: Overall performance (averaged accuracy) under the non-label-sharing scenario. MLTI outperforms other strategies and improves the generalization ability.

| Backbone | Strategies | miniImagenet-S | | ISIC | | DermNet-S | | Tabular Murris | |
|---|---|---|---|---|---|---|---|---|---|
| | | 1-shot | 5-shot | 1-shot | 5-shot | 1-shot | 5-shot | 1-shot | 5-shot |
| MAML | Vanilla | 38.27% | 52.14% | 57.59% | 65.24% | 43.47% | 60.56% | 79.08% | 88.55% |
| | Meta-Reg | 38.35% | 51.74% | 58.57% | 68.45% | 45.01% | 60.92% | 79.18% | 89.08% |
| | TAML | 38.70% | 52.75% | 58.39% | 66.09% | 45.73% | 61.14% | 79.82% | 89.11% |
| | Meta-Dropout | 38.32% | 52.53% | 58.40% | 67.32% | 44.30% | 60.86% | 78.18% | 89.25% |
| | MetaMix | 39.43% | 54.14% | 60.34% | 69.47% | 46.81% | 63.52% | 81.06% | 89.75% |
| | Meta-Maxup | 39.28% | 53.02% | 58.68% | 69.16% | 46.10% | 62.64% | 79.56% | 88.88% |
| | **MLTI (ours)** | **41.58%** | **55.22%** | **61.79%** | **70.69%** | **48.03%** | **64.55%** | **81.73%** | **91.08%** |
| ProtoNet | Vanilla | 36.26% | 50.72% | 58.56% | 66.25% | 44.21% | 60.33% | 80.03% | 89.20% |
| | MetaMix | 39.67% | 53.10% | 60.58% | 70.12% | 47.71% | 62.68% | 80.72% | 89.30% |
| | Meta-Maxup | 39.80% | 53.35% | 59.66% | 68.97% | 46.06% | 62.97% | 80.87% | 89.42% |
| | **MLTI (ours)** | **41.36%** | **55.34%** | **62.82%** | **71.52%** | **49.38%** | **65.19%** | **81.89%** | **90.12%** |

respectively. More details on the datasets and set-up are provided in Appendix D.1. We adopt MSE to measure the performance for the Pose regression dataset and accuracy for the classification datasets.

**Results.** Under the label-sharing scenario, we report the overall performance and analyze the compatibility of MLTI in Table 1 and Appendix D.2, respectively. According to Table 1, we observe that MLTI outperforms other regularization strategies across the board, including passively adding regularization (i.e., Meta-Reg, TAML, Meta-Dropout) and augmenting tasks individually (i.e., Meta-Aug, MetaMix, Meta-Maxup). These results indicate that MLTI consistently improves generalization through interpolation on the task distribution. The claim is further be strengthened by the compatibility analysis (Appendix D.2), where MLTI boosts the performance of a variety of meta-learning algorithms. We also investigate the effect of the number of meta-training tasks and report the performance in Appendix D.3. We observe that the improvements from MLTI are robust under different settings but that the greatest improvements come when the number of tasks is limited.

**Ablation study.** In Table 2, we conduct an ablation study under the label-sharing scenario. Here, we investigate how MLTI performs compared with only applying intra-task interpolation (i.e., $\mathcal{T}_i = \mathcal{T}_j$) and cross-task interpolation (i.e., $\mathcal{T}_i \neq \mathcal{T}_j$), which are denoted as Intra-Intrpl and Cross-Intrpl, respectively. We observe that both Intra-Intrpl and Cross-Intrpl outperform the vanilla approach without task augmentation and that MLTI achieves the best performance, indicating that the strategies are complementary to some degree. In addition, cross-interpolation outperforms the intra-interpolation in most datasets. The results corroborate the effectiveness of cross-task interpolation when tasks are sparsely sampled from the data distribution.

## 6.2 NON-LABEL-SHARING SCENARIO

**Datasets and experimental setup.** Under the non-label-sharing scenario, we conduct experiments on four datasets: (1) general image classification on miniImagenet (Vinyals et al., 2016); (2)&(3) medical image classification on ISIC (Milton, 2019) and DermNet (Der, 2016); and (4) cell type classification across organs on Tabular Murris (Cao et al., 2021). Since a task in meta-learning is defined to correspond to a particular data-generating distribution (Finn et al., 2017; Rajeswaran et al., 2019), the number of distinct meta-training tasks in $N$-way classification is actually the number of ways to choose $N$ from all base classes. Thus, for miniImagenet and Dermnet, we reduce the number tasks by limiting the number of meta-training classes (a.k.a., base classes) and obtain the *miniImagenet-S*, *ISIC*, *DermNet-S*, *Tabular Murris* benchmarks, whose base classes are 12, 4, 30, 57, respectively (see the experiments on full-size miniImagenet and DermNet in Appendix E.2). The experiments are performed under the $N$-way $K$-shot setting (Finn and Levine, 2018), where $N = 2$ for ISIC and $N = 5$ for the rest datasets. Note that, Meta-Aug (Rajendran et al., 2020) under the non-label-sharing scenario is exactly the same as the label shuffling, which is already adopted in vanilla MAML and ProtoNet. Due to space limitations, we report only the accuracy for the non-label-sharing scenario here and provide the full table with 95% confidence intervals in Appendix E.9. More details about the datasets and set-up are in Appendix E.1.

**Results.** Table 3 gives the results of MLTI and prior methods. MLTI consistently outperforms other strategies. The performance gains suggest that MLTI can improve the generalization ability of meta-learning amidst sparsely sampled task distributions. We also analyze of compatibility of MLTI under the non-label-sharing scenario in Table 9 of Appendix E.3. The results validate that MLTI can robustly boost performance with different backbone methods. For the ablation study, we repeat

Table 4: Cross-domain adaptation under the non-label-sharing scenario. A → B represents that the model is meta-trained on A and then is meta-tested on B.

| Model | | mini → Dermnet | | Dermnet → mini | |
|---|---|---|---|---|---|
| | | 1-shot | 5-shot | 1-shot | 5-shot |
| MAML | | 33.67% | 50.40% | 28.40% | 40.93% |
| | +MLTI | **36.74%** | **52.56%** | **30.03%** | **42.25%** |
| ProtoNet | | 33.12% | 50.13% | 28.11% | 40.35% |
| | +MLTI | **35.46%** | **51.79%** | **30.06%** | **42.23%** |

the experiments on the non-label-sharing scenario and report the results in Table 10 of Appendix E.4. We additionally provide base model and hyperparameter analysis in Appendix E.5. MLTI achieves the best performance across various settings, further strengthen its effectiveness.

**Cross-domain adaptation.** To further evaluate performance of MLTI, we conduct a comparison under the cross-domain adaptation setting where we meta-train the model on one source domain and evaluate it on another target domain. We perform cross-domain adaptation across miniImagenet-S and Dermnet-S and report the performance under MAML and ProtoNet in Table 4. The results validate that MLTI can improve generalization even in this more challenging setting.

**Effect of the number of meta-training tasks.**
We analyze the effect of the number of tasks under 5-shot setting (with a ProtoNet backbone) in Figures 2a and 2b (see more results in Appendix E.6). We have two key observations: (1) MLTI consistently improves the performance for all numbers of tasks, showing its effectiveness and robustness; (2) The improvement gap between MLTI and the vanilla model decreases as the the number of tasks increases on miniImagenet, and keeps consistent on DermNet. We expect this is because the meta-training tasks may be more related to meta-testing tasks in miniImagenet, than in DermNet. Besides, we

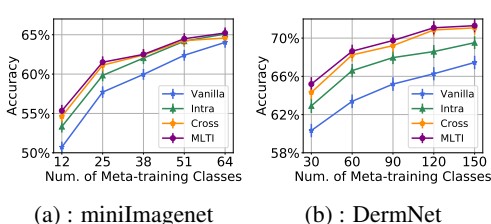

(a) : miniImagenet  (b) : DermNet

Figure 2: Accuracy w.r.t. the num. of tasks under the non-label-sharing scenario. Intra and Cross represent intra-task interpolation (i.e., $\mathcal{T}_i = \mathcal{T}_j$) and cross-task interpolation (i.e., $\mathcal{T}_i \neq \mathcal{T}_j$).

conduct an additional experiments in Appendix E.7 to show the promise of MLTI when we only have extremely limited tasks.

**Analysis of Interpolated Tasks.** Building upon ProtoNet, we show the t-SNE (Maaten and Hinton, 2008) visualization of both original tasks and interpolated tasks in Figure 3. Specifically, we randomly select 3 original tasks and 300 interpolated tasks under the 1-shot miniImagenet-S setting, where the color of each interpolated task indicates its proximity to the corresponding original tasks. Each task is represented by the averaged representation over its corresponding prototypes, where we combine both support and query sets to calculate the prototypes. The figure suggests that the interpolated tasks generated by MLTI indeed densify the task distribution and bridge the gap between different tasks.

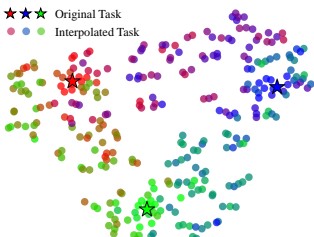

Figure 3: Visualization of the original and interpolated tasks.

## 7 CONCLUSION

In this paper, we investigate the problem of meta-learning with fewer tasks and propose a new task interpolation strategy MLTI. The proposed MLTI targets the task distribution directly to generate more meta-training tasks via task interpolation for both label-sharing and non-label-sharing scenarios. The consistent performance gains across eight datasets demonstrate that MLTI improves the generalization of meta-learning algorithms especially when the number of available meta-training tasks is small, which is further supported by the theoretical analysis.

## REPRODUCIBILITY STATEMENT

For our theoretical results, a complete proof of all claims and the discussion of assumptions are provided in Appendix B. For our empirical results, we discuss the details of datasets and list all hyperparameters under the label-sharing scenario and non-label-sharing scenario in Appendix D.1 and E.1, respectively. Code: https://github.com/huaxiuyao/MLTI.

## ACKNOWLEDGEMENT

This work was supported in part by JPMorgan Chase & Co and Juniper Networks. Any views or opinions expressed herein are solely those of the authors listed, and may differ from the views and opinions expressed by JPMorgan Chase & Co., Juniper Networks or their affiliates. This material is not a product of the Research Department of J.P. Morgan Securities LLC and Juniper Networks. This material should not be construed as an individual recommendation for any particular client and is not intended as a recommendation of particular securities, financial instruments or strategies for a particular client. This material does not constitute a solicitation or offer in any jurisdiction. Linjun Zhang would like to acknowledge the support from NSF DMS-2015378.

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

## A   PSEUDOCODES

In this section, we show the pseudocodes for MLTI with MAML (meta-training process: Alg. 1, meta-testing process: Alg. 2) and ProtoNet (meta-training process: Alg. 3, meta-testing process: Alg. 4).

---

**Algorithm 1** Meta-training Process of MAML with MLTI

---

**Require:** $p(\mathcal{T})$: task distribution; $\eta, \gamma$: inner- and outer-loop learning rate; $L^s$: the number of shared layers; Beta distribution
 1: Randomly initialize the model initial parameters $\theta$
 2: **while** not converge **do**
 3:     Randomly sample a batch of tasks $\{\mathcal{T}_i\}_{i=1}^{|I|}$ with dataset
 4:     **for** each task $\mathcal{T}_i$ **do**
 5:         Sample a support set $\mathcal{D}_i^s = (\mathbf{X}_i^s, \mathbf{Y}_i^s)$ and a query set $\mathcal{D}_i^q = (\mathbf{X}_i^q, \mathbf{Y}_i^q)$ from $\mathcal{D}_i$
 6:         Sample another task $\mathcal{T}_j$ (allow $i = j$) from $\{\mathcal{T}_i\}_{i=1}^{|I|}$ with corresponding support set $\mathcal{D}_j^s = (\mathbf{X}_j^s, \mathbf{Y}_j^s)$ and query set $\mathcal{D}_j^q = (\mathbf{X}_j^q, \mathbf{Y}_j^q)$
 7:         Random sample one layer $l$ from the shared layers
 8:         Obtain the hidden representations $\mathbf{H}_i^{s,l}, \mathbf{H}_i^{q,l}, \mathbf{H}_j^{s,l}, \mathbf{H}_j^{q,l}$ of the support/query sets of task $\mathcal{T}_i$ and $\mathcal{T}_j$
 9:         Apply task interpolation between task $\mathcal{T}_i$ and $\mathcal{T}_j$ via Eqn. (5) (label-sharing tasks) or Eqn. (6) (non-label-sharing tasks), and obtain the interpolated support set $\mathcal{D}_{i,cr}^s = (\tilde{\mathbf{H}}_{i,cr}^{s,l}, \tilde{\mathbf{Y}}_{i,cr}^s)$ and query set $\mathcal{D}_{i,cr}^q = (\tilde{\mathbf{H}}_{i,cr}^{q,l}, \tilde{\mathbf{Y}}_{i,cr}^q)$
10:         Calculate the task-specific parameters $\phi_{i,cr}^{L-l}$ by the inner-loop adaptation, i.e., $\phi_{i,cr}^{L-l} = \theta^{L-l} - \eta\nabla_{\theta^{L-l}}\mathcal{L}(f_{\theta^{L-l}}^{MAML}; \mathcal{D}_{i,cr}^s)$
11:     **end for**
12:     Optimize the model initial parameters as $\theta \leftarrow \theta - \gamma\frac{1}{|I|}\sum_{i=1}^{|I|}\mathcal{L}(f_{\phi_{i,cr}^{L-l}}^{MAML}; \mathcal{D}_{i,cr}^q)$
13: **end while**

---

---

**Algorithm 2** Meta-testing Process of MAML with MLTI

---

**Require:** $p(\mathcal{T})$: task distribution; $\eta$: inner-loop learning rate; $\theta^*$: learned model initial parameters
 1: Randomly initialize the model initial parameters $\theta$
 2: **for** each task $\mathcal{T}_t$ with support set $\mathcal{D}_t^s$ and query set $\mathcal{D}_t^q$ **do**
 3:     Calculate the task-specific parameters $\phi_i$ by the inner-loop adaptation, i.e., $\phi_i = \theta^* - \eta\nabla_{\theta^*}\mathcal{L}(f_{\theta^*}^{MAML}; \mathcal{D}_i^s)$
 4:     Obtain the predicted labels of the query set by $f_{\phi_i}^{MAML}(\mathcal{D}_i^q)$ and evaluate the performance
 5: **end for**

---

## B   ADDITIONAL THEORETICAL ANALYSIS

### B.1   PROOFS OF NON-LABEL-SHARING SCENARIO

#### B.1.1   PROOF OF LEMMA 1

*Proof.* Recall that the interpolated dataset is $\mathcal{D}_{i,cr}^q = (\tilde{\mathbf{H}}_{i,cr}^{q,1}, \tilde{\mathbf{Y}}_{i,cr}^q) := \{(\mathbf{h}_{i,k,cr}^1, \mathbf{y}_{i,k,cr})\}_{k=1}^N$, where

$$\mathbf{h}_{i,k,cr;r}^1 = \lambda\mathbf{h}_{i,k;r}^1 + (1-\lambda)\mathbf{h}_{j,k';r'}^1, \quad \mathbf{y}_{i,k,cr} = Lb(r, r').$$

---

**Algorithm 3** Meta-training Process of ProtoNet with MLTI

---

**Require:** $p(\mathcal{T})$: task distribution; $\gamma$: learning rate; Beta distribution
 1: Randomly initialize the model initial parameters $\theta$
 2: **while** not converge **do**
 3:     Randomly sample a batch of tasks $\{\mathcal{T}_i\}_{i=1}^{|I|}$ with dataset
 4:     **for** each task $\mathcal{T}_i$ **do**
 5:         Sample a support set $\mathcal{D}_i^s = (\mathbf{X}_i^s, \mathbf{Y}_i^s)$ and a query set $\mathcal{D}_i^q = (\mathbf{X}_i^q, \mathbf{Y}_i^q)$ from $\mathcal{D}_i$
 6:         Sample another task $\mathcal{T}_j$ (allow $i = j$) from $\{\mathcal{T}_i\}_{i=1}^{|I|}$ with corresponding support set $\mathcal{D}_j^s = (\mathbf{X}_j^s, \mathbf{Y}_j^s)$ and query set $\mathcal{D}_j^q = (\mathbf{X}_j^q, \mathbf{Y}_j^q)$
 7:         Random sample one layer $l$ from the shared layers
 8:         Obtain the hidden representations $\mathbf{H}_i^{s,l}$, $\mathbf{H}_i^{q,l}$, $\mathbf{H}_j^{s,l}$, $\mathbf{H}_j^{q,l}$ of the support/query sets of task $\mathcal{T}_i$ and $\mathcal{T}_j$
 9:         Apply task interpolation between task $\mathcal{T}_i$ and $\mathcal{T}_j$, and obtain the interpolated support set $\mathcal{D}_{i,cr}^s = (\tilde{\mathbf{H}}_{i,cr}^{s,l}, \tilde{\mathbf{Y}}_{i,cr}^s)$ and query set $\mathcal{D}_{i,cr}^q = (\tilde{\mathbf{H}}_{i,cr}^{q,l}, \tilde{\mathbf{Y}}_{i,cr}^q)$
10:         Calculate the prototypes $\{\mathbf{c}_r\}_{r=1}^R$ ($N_r$ represents the number of samples in class $r$) by $\mathbf{c}_r = \frac{1}{N_r} \sum_{(\mathbf{h}_{i,k,cr;r}^s, \mathbf{y}_{i,k,cr;r}^s) \in \mathcal{D}_{i,cr;r}^s} f_{\theta^{L-l}}^{PN}(\mathbf{h}_{i,k,cr;r}^s)$
11:         Calculate the loss of task $\mathcal{T}_i$ as $\mathcal{L}_i = -\sum_k \log \frac{\exp(-d(f_{\theta^{L-l}}^{PN}(\mathbf{h}_{i,k,cr}^q), \mathbf{c}_r))}{\sum_{r'} \exp(-d(f_{\theta^{L-l}}^{PN}(\mathbf{h}_{i,k,cr}^q), \mathbf{c}_{r'}))}$
12:     **end for**
13:     Update $\theta \leftarrow \theta - \gamma \frac{1}{|I|} \sum_{i=1}^{|I|} \mathcal{L}_i$
14: **end while**

---

**Algorithm 4** Meta-testing Process of ProtoNet with MLTI

---

**Require:** $p(\mathcal{T})$: task distribution; $\theta^*$: learned parameter of the base model
 1: **for** each task $\mathcal{T}_t$ with support set $\mathcal{D}_t^s$ and query set $\mathcal{D}_t^q$ **do**
 2:     Calculate the prototypes $\{\mathbf{c}_r\}_{r=1}^R$ ($N_r$ represents the number of samples in class $r$) by $\mathbf{c}_r = \frac{1}{N_r} \sum_{(\mathbf{h}_{i,k;r}^s, \mathbf{y}_{i,k;r}^s) \in \mathcal{D}_{i;r}^s} f_\theta^{PN}(\mathbf{h}_{i,k;r}^s)$
 3:     Calculate the probability of each sample being assigned to class $r$ as $p(\mathbf{y}_{i,k}^q = r | \mathbf{x}_{i,k}^q) = \frac{\exp(-d(f_\theta^{PN}(\mathbf{h}_{i,k,cr}^q), \mathbf{c}_r))}{\sum_{r'} \exp(-d(f_\theta^{PN}(\mathbf{h}_{i,k,cr}^q), \mathbf{c}_{r'}))}$
 4:     Obtain the predicted class as $\hat{\mathbf{y}}_{i,k}^q = \arg\max_r p(\mathbf{y}_{i,k}^q = r | \mathbf{x}_{i,k}^q)$ and evaluate the performance
 5: **end for**

---

Here, $r = \mathbf{y}_{i,k}$, $\lambda \sim \text{Beta}(\alpha, \beta)$, $j \sim U([|I|])$, $r \sim U([R_i])$, where $R_i$ represents the number of classes in task $\mathcal{T}_i$, and $Lb(r, r')$ denotes the label uniquely determined by the pair $(r, r')$. The superscript $q$ is also omitted in the whole section. Since for a give set of $r'$, $r$ and $(r, r')$ has a one-to-one correspondence, without loss of generality, we assume $r = (r, r')$ in this classification setting.

Recall that $\mathcal{L}_t(\{\mathcal{D}_{i,cr}\}_{i=1}^{|I|}) = \frac{1}{|I|} \sum_{i=1}^{|I|} \mathcal{L}(\mathcal{D}_{i,cr}) = \frac{1}{|I|} \sum_{i=1}^{|I|} \frac{1}{N} \sum_{k=1}^N \mathcal{L}(f_{\phi_i}(\mathbf{x}_{i,k,cr}), \mathbf{y}_{i,k,cr}) = \frac{1}{|I|} \sum_{i=1}^n \frac{1}{N} \sum_{k=1}^N \mathcal{L}(\mathbf{h}_{i,k,cr}, \mathbf{y}_{i,k,cr})$. Then let us compute the second-order Taylor expansion on $\mathcal{L}_t(\{\mathcal{D}_{i,cr}\}_{i=1}^{|I|}) = \frac{1}{|I|} \sum_{i=1}^{|I|} \frac{1}{N} \sum_{k=1}^N \mathcal{L}(\mathbf{h}_{i,k,cr}^1, \mathbf{y}_{i,k,cr})$ with respect to the first argument around $\frac{1}{\lambda} \mathbb{E}[\mathbf{h}_{i,k,cr}^1 \mid \mathbf{h}_{i,k}^1] = \mathbf{h}_{i,k,cr}^1$, we have the Taylor expansion of $\mathcal{L}_t(\{\mathcal{D}_{i,cr}\}_{i=1}^{|I|})$ up to the second-order equals to

$$\frac{1}{|I|} \sum_{i=1}^{|I|} \mathcal{L}(\bar{\lambda} \mathcal{D}_i) + c \frac{1}{|I|} \sum_{i=1}^{|I|} \frac{1}{N} \sum_{k=1}^N \psi(\mathbf{h}_{i,k;r}^{1\top} \phi_i) \phi_i^\top Cov(\mathbf{h}_{i,k,cr}^1 \mid \mathbf{h}_{i,k}^1) \phi_i \tag{12}$$

$$= \frac{1}{|I|} \sum_{i=1}^{|I|} \mathcal{L}(\bar{\lambda} \mathcal{D}_i) + c \frac{1}{|I|} \sum_{i=1}^{|I|} \frac{1}{N} \sum_{k=1}^N \psi(\mathbf{h}_{i,k;r}^{1\top} \phi_i) \cdot \phi_i^\top \left( \frac{1}{|I|} \sum_{i=1}^{|I|} \frac{1}{2} \sum_{r=1}^2 \frac{1}{N_{i,r}} \sum_{k=1}^{N_{i,r}} \mathbf{h}_{i,k;r}^1 \mathbf{h}_{i,k;r}^{1\top} \right) \phi_i \tag{13}$$

$$= \mathcal{L}_t(\bar{\lambda} \{\mathcal{D}_i\}_{i=1}^{|I|}) + c \frac{1}{N|I|} \sum_{i=1}^{|I|} \sum_{k=1}^N \psi(\mathbf{h}_{i,k;r}^{1\top} \phi_i) \cdot \phi_i^\top \left( \frac{1}{|I|} \sum_{i=1}^{|I|} \frac{1}{2} \sum_{r=1}^2 \frac{1}{N_{i,r}} \sum_{i=1}^{|I|} \sum_{k=1}^{N_{i,r}} \mathbf{h}_{i,k;r}^1 \mathbf{h}_{i,k;r}^{1\top} \right) \phi_i,$$

where $c = \mathbb{E}[\frac{(1-\lambda)^2}{\lambda^2}]$ and the second equality (13) uses the fact that the data is pre-processed so that $\frac{1}{|I|} \sum_{i=1}^{|I|} \frac{1}{2} \sum_{r=1}^{2} \frac{1}{N_{i,r}} \sum_{i=1}^{|I|} \sum_{k=1}^{N_{i,r}} \mathbf{h}_{i,k;r} = 0$.  $\square$

### B.1.2 PROOF OF THEOREM 1

We first state a standard uniform deviation bound based on Rademacher complexity (c.f. (Bartlett and Mendelson, 2002)).

**Lemma 3.** *Assume $\{z_1, ..., z_N\}$ are drawn i.i.d. from a distribution $P$ over $\mathcal{Z}$, and $\mathcal{G}$ denotes function class on $\mathcal{Z}$ with members mapping from $\mathcal{Z}$ to $[a, b]$. With probability at least $1 - \delta$ over the draw of the sample and $\delta > 0$, we have the following bound:*

$$\sup_{g \sim \mathcal{G}} \|\mathbb{E}_{\hat{P}} g(z) - \mathbb{E}_P g(z)\| \leq 2R(\mathcal{G}; z_1, ..., z_N) + \sqrt{\frac{\log(1/\delta)}{N}},$$

*where $R(\mathcal{G}; z_1, ..., z_N)$ represents the Rademacher complexity of the function class $\mathcal{G}$.*

*Proof.* We now formulate $\mathcal{R}(\{\mathcal{D}_i\}_{i=1}^{|I|}) - \mathcal{R}$ as

$$
\begin{aligned}
\mathcal{R}(\{\mathcal{D}_i\}_{i=1}^{|I|}) - \mathcal{R} =& \mathbb{E}_{\mathcal{T}_i \sim \hat{p}(\mathcal{T})} \mathbb{E}_{(\mathbf{X}_i, \mathbf{Y}_i) \sim \hat{p}(\mathcal{T}_i)} \mathcal{L}(f_{\phi_i}(\mathbf{X}_i), \mathbf{Y}_i) - \mathbb{E}_{\mathcal{T}_i \sim p(\mathcal{T})} \mathbb{E}_{(\mathbf{X}_i, \mathbf{Y}_i) \sim \mathcal{T}_i} [\mathcal{L}(f_{\phi_i}(\mathbf{X}_i), \mathbf{Y}_i)] \\
=& \underbrace{\mathbb{E}_{\mathcal{T}_i \sim \hat{p}(\mathcal{T})} \mathbb{E}_{(\mathbf{X}_i, \mathbf{Y}_i) \sim \hat{p}(\mathcal{T}_i)} \mathcal{L}(f_{\phi_i}(\mathbf{X}_i), \mathbf{Y}_i) - \mathbb{E}_{\mathcal{T}_i \sim \hat{p}(\mathcal{T})} \mathbb{E}_{(\mathbf{X}_i, \mathbf{Y}_i) \sim \mathcal{T}_i} [\mathcal{L}(f_{\phi_i}(\mathbf{X}_i), \mathbf{Y}_i)]}_{(i)} \\
& + \underbrace{\mathbb{E}_{\mathcal{T}_i \sim \hat{p}(\mathcal{T})} \mathbb{E}_{(\mathbf{X}_i, \mathbf{Y}_i) \sim \mathcal{T}_i} \mathcal{L}(f_{\phi_i}(\mathbf{X}_i), \mathbf{Y}_i) - \mathbb{E}_{\mathcal{T}_i \sim p(\mathcal{T})} \mathbb{E}_{(\mathbf{X}_i, \mathbf{Y}_i) \sim \mathcal{T}_i} [\mathcal{L}(f_{\phi_i}(\mathbf{X}_i), \mathbf{Y}_i)]}_{(ii)}.
\end{aligned}
\tag{14}
$$

Recall that we consider the function $f_{\phi_i}^{MAML}(\mathbf{X}_i) = \phi_i^\top \sigma(\mathbf{W} \mathbf{X}_i) := \phi_i^\top \mathbf{H}_i^1$ and the function class

$$\mathcal{F}_\gamma = \{\mathbf{H}^{1\top} \phi : \mathbb{E}[\psi(\mathbf{H}^{1\top} \phi)] \phi^\top \Sigma \phi \leq \gamma\}.$$

For each $\mathcal{T}_i$, let us consider $f_{\phi_i}(\cdot) \in \mathcal{F}_\gamma$. Combining Theorem 3.4 and Theorem A.1 in Zhang et al. (2021), we have the following result for the Rademacher complexity:

$$
\begin{aligned}
R(\mathcal{F}_\gamma; z_1, ..., z_n) \leq& 2 \max\{(\frac{\gamma}{\rho})^{1/4}, (\frac{\gamma}{\rho})^{1/2}\} \sqrt{\frac{(rank(\Sigma_{\sigma, \mathcal{T}}) + \|\Sigma_{\mathcal{T}}^{\mathbf{W}\dagger/2} \mu_{\sigma, \mathcal{T}}\|)}{N}} \\
\leq& 2 \max\{(\frac{\gamma}{\rho})^{1/4}, (\frac{\gamma}{\rho})^{1/2}\} \cdot \sqrt{\frac{R + U}{N}}.
\end{aligned}
\tag{15}
$$

Then, we bound the first term (i) in Eqn. (14) can be as below.

$$
\begin{aligned}
& \mathbb{E}_{\mathcal{T}_i \sim \hat{p}(\mathcal{T})} \mathbb{E}_{(\mathbf{X}_i, \mathbf{Y}_i) \sim \hat{p}(T_i)} \mathcal{L}(f_{\phi_i}(\mathbf{X}_i), \mathbf{Y}_i) - \mathbb{E}_{\mathcal{T}_i \sim \hat{p}(\mathcal{T})} \mathbb{E}_{(\mathbf{X}_i, \mathbf{Y}_i) \sim \mathcal{T}_i} [\mathcal{L}(f_{\phi_i}(\mathbf{X}_i), \mathbf{Y}_i)] \\
\leq& \mathbb{E}_{T_i \sim \hat{p}(\mathcal{T})} |\mathbb{E}_{(\mathbf{X}_i, \mathbf{Y}_i) \sim \hat{p}(T_i)} \mathcal{L}(f_{\phi_i}(\mathbf{X}_i), \mathbf{Y}_i) - \mathbb{E}_{(\mathbf{X}_i, \mathbf{Y}_i) \sim \mathcal{T}_i} [\mathcal{L}(f_{\phi_i}(\mathbf{X}_i), \mathbf{Y}_i)] \\
\leq& C_1 \max\{(\frac{\gamma}{\rho})^{1/4}, (\frac{\gamma}{\rho})^{1/2}\} \sqrt{\frac{(R + U)}{N}} + C_2 \sqrt{\frac{\log(|I|/\delta)}{N}},
\end{aligned}
$$

where $C_1$ and $C_2$ are constants, and the additional $\log(|I|)$ term in the last inequality above is caused by taking the union bound on $|I|$ tasks.

Denote function $g : \mathcal{T} \to \mathbb{R}$ such that $g(\mathcal{T}) = \mathbb{E}_{(\mathbf{X}, \mathbf{Y}) \sim \mathcal{D}}(\mathcal{L}(f_\phi(\mathbf{X}), \mathbf{Y}))$. Denote

$$\mathcal{G} = \{g(\mathcal{T}) : g(\mathcal{T}) = \mathbb{E}_{(\mathbf{X}, \mathbf{Y}) \sim \mathcal{D}}(\mathcal{L}(f_\phi(\mathbf{X}), \mathbf{Y})), f_\phi \in \mathcal{F}_\gamma\}.$$

Let $A(x) = 1/(1 + e^x)$. The second term (ii) in Eqn. (14) requires computing the Rademacher complexity for the function class over distributions

$$
\begin{aligned}
R(\mathcal{G}; \mathcal{T}_1, ..., \mathcal{T}_{|I|}) =& \mathbb{E} \sup_{g \in \mathcal{G}} \frac{1}{|I|} | \sum_{i=1}^{|I|} \sigma_i g(\mathcal{T}_i)| = \mathbb{E} \sup_{g \in \mathcal{G}} \frac{1}{|I|} | \sum_{i=1}^{|I|} \sigma_i \mathbb{E}_{(\mathbf{X},\mathbf{Y}) \sim \mathcal{T}_i} (A(f_{\phi_i}(\mathbf{X})) - \mathbf{XY}| \\
\lesssim& \mathbb{E} \sup_{g \in \mathcal{G}} \frac{1}{|I|} | \sum_{i=1}^{|I|} \sigma_i \mathbb{E}_{(\mathbf{X},\mathbf{Y}) \sim \mathcal{T}_i} f_{\phi_i}(\mathbf{X})| + \mathbb{E} \sup_{g \in \mathcal{G}} \frac{1}{|I|} | \sum_{i=1}^{|I|} \sigma_i \mathbb{E}_{(\mathbf{X},\mathbf{Y}) \sim \mathcal{T}_i} \mathbf{Y}| \\
\leq& \mathbb{E} \sup_{g \in G} \frac{1}{|I|} | \sum_{i=1}^{|I|} \sigma_i (\Sigma^{1/2} \phi_i)^\top \Sigma^{\dagger/2} \mu_{\sigma,\mathcal{T}}| + \sqrt{\frac{1}{|I|}} \\
\leq& \max\{(\frac{\gamma}{\rho})^{1/4}, (\frac{\gamma}{\rho})^{1/2}\} \sqrt{\frac{R+U}{|I|}} + \sqrt{\frac{1}{|I|}}.
\end{aligned}
$$

Then we have the following bound on (ii):

$$
\begin{aligned}
&\mathbb{E}_{\mathcal{T}_i \sim \hat{p}(\mathcal{T})} \mathbb{E}_{(\mathbf{X}_i, \mathbf{Y}_i) \sim \mathcal{T}_i} \mathcal{L}(f_{\phi_i}(\mathbf{X}_i), \mathbf{Y}_i) - \mathbb{E}_{\mathcal{T}_i \sim p(\mathcal{T})} \mathbb{E}_{(\mathbf{X}_i, \mathbf{Y}_i) \sim \mathcal{T}_i} [\mathcal{L}(f_{\phi_i}(\mathbf{X}_i), \mathbf{Y}_i)] \\
\leq& C_3 \max\{(\frac{\gamma}{\rho})^{1/4}, (\frac{\gamma}{\rho})^{1/2} \sqrt{\frac{U}{|I|}} + C_4 \sqrt{\frac{\log(1/\delta)}{|I|}}.
\end{aligned}
$$

Combining the pieces, we obtain the desired result. With probability at least $1 - \delta$,

$$
\begin{aligned}
|\mathcal{R}(\{\mathcal{D}_i\}_{i=1}^{|I|}) - \mathcal{R}| \leq& A_1 \max\{(\frac{\gamma}{\rho})^{1/4}, (\frac{\gamma}{\rho})^{1/2}\}(\sqrt{\frac{R+U}{N}} + \sqrt{\frac{R+U}{|I|}}) \\
&+ A_2 \sqrt{\frac{\log(|I|/\delta)}{N}} + A_3 \sqrt{\frac{\log(1/\delta)}{|I|}}.
\end{aligned}
$$

$\square$

### B.1.3 PROOF OF LEMMA 2

Recall that we apply MLTI in the feature space for theoretical analysis, the interpolated dataset is then denoted as $\mathcal{D}_{i,cr}^q = (\tilde{\mathbf{X}}_{i,cr}^q, \tilde{\mathbf{Y}}_{i,cr}^q) := \{(\mathbf{x}_{i,k,cr}, \mathbf{y}_{i,k,cr})\}_{k=1}^N$, where

$$
\mathbf{x}_{i,k,cr;r} = \lambda \mathbf{x}_{i,k;r} + (1 - \lambda)\mathbf{x}_{j,k';r'}, \quad \mathbf{y}_{i,k,cr} = Lb(r, r').
$$

where $r = \mathbf{y}_{i,k}, \lambda \sim \text{Beta}(\alpha, \beta), j \sim U([|I|]), r \sim U([2])$, and $Lb(r, r')$ denotes the label uniquely determined by the pair $(r, r')$. Since for a give set of $r', r$ and $(r, r')$ has a one-to-one correspondence, without loss of generality, we assume $r = (r, r')$ in this classification setting.

*Proof.* To prove Lemma 2, first, we would like to note that since the overall sample mean $\frac{1}{|I|} \sum_{i=1}^{|I|} \frac{1}{2} \sum_{r=1}^2 \frac{1}{N_{i,r}} \sum_{k=1}^{N_{i,r}} \mathbf{x}_{i,k;r} = 0$, we then have

$$
\mathbb{E}[\mathbf{x}_{i,k,cr;r} \mid \mathbf{x}_{i,k;r}] = \mathbf{x}_{i,k;r}.
$$

Then let us compute the second-order Taylor expansion on $\mathcal{L}_t(\{\mathcal{D}_{i,cr}\}_{i=1}^{|I|}) = \frac{1}{|I|} \sum_{i=1}^{|I|} \frac{1}{N} \sum_{k=1}^N \mathcal{L}(\mathbf{x}_{i,k,cr}, \mathbf{y}_{i,k,cr}) = (N|I|)^{-1} \sum_{i,k} (1 + \exp(\langle(\mathbf{x}_{i,k,cr} - (\mathbf{c}_{1,cr} + \mathbf{c}_{2,cr})/2, \theta\rangle))^{-1}$ with respect to the first argument around $\frac{1}{\lambda} \mathbb{E}[\mathbf{x}_{i,k,cr} \mid \mathbf{x}_{i,k}] = \mathbf{x}_{i,k,cr}$, we have that the Taylor

expansion of $\mathcal{L}_t(\{\mathcal{D}_{i,cr}\}_{i=1}^{|I|})$ up to the second-order equals to

$$
\frac{1}{|I|}\sum_{i=1}^{|I|}\mathcal{L}(\bar{\lambda}\mathcal{D}_i) + c\frac{1}{|I|}\sum_{i=1}^{|I|}\frac{1}{N}\sum_{k=1}^{N}\psi(\mathbf{x}_{i,k}^{\top}\theta)\theta^{\top}Cov(\mathbf{x}_{i,k,cr} \mid \mathbf{x}_{i,k})\theta
$$

$$
=\frac{1}{|I|}\sum_{i=1}^{|I|}\mathcal{L}(\bar{\lambda}\mathcal{D}_i) + c\frac{1}{|I|}\sum_{i=1}^{|I|}\frac{1}{N}\sum_{k=1}^{N}\psi(\mathbf{x}_{i,k}^{\top}\theta)\cdot\theta^{\top}(\frac{1}{|I|}\sum_{i=1}^{|I|}\frac{1}{2}\sum_{r=1}^{2}\frac{1}{N_r}\sum_{k=1}^{N_r}\mathbf{x}_{i,k;r}\mathbf{x}_{i,k;r}^{\top})\theta
$$

$$
=\mathcal{L}_t(\bar{\lambda}\{\mathcal{D}_i\}_{i=1}^{|I|}) + c\frac{1}{N|I|}\sum_{i=1}^{|I|}\sum_{k=1}^{N}\psi(\mathbf{x}_{i,k}^{\top}\theta)\cdot\theta^{\top}(\frac{1}{|I|}\sum_{i=1}^{|I|}\frac{1}{2}\sum_{r=1}^{2}\frac{1}{N_r}\sum_{i=1}^{|I|}\sum_{k=1}^{N_r}\mathbf{x}_{i,k;r}\mathbf{x}_{i,k;r}^{\top})\theta,
$$

$$
=\mathcal{L}_t(\bar{\lambda}\{\mathcal{D}_i\}_{i=1}^{|I|})
$$

$$
+ c\frac{1}{N|I|}\sum_{i\in I,k\in[N]}\psi(\langle\mathbf{x}_{i,k}-(\mathbf{c}_1+\mathbf{c}_2)/2,\theta\rangle)\cdot\theta^{\top}(\frac{1}{|I|}\sum_{i=1}^{|I|}\frac{1}{2}\sum_{r=1}^{2}\frac{1}{N_r}\sum_{i=1}^{|I|}\sum_{k=1}^{N_r}\mathbf{x}_{i,k;r}\mathbf{x}_{i,k;r}^{\top})\theta
$$

where $c = \mathbb{E}[\frac{(1-\lambda)^2}{\lambda^2}]$. $\qquad\qquad\square$

### B.1.4 PROOF OF THEOREM 2

Similar to the proof of Theorem 1, we use Lemma 3 in the proof of Theorem 2.

*Proof.* We first write $\mathcal{R}(\{\mathcal{D}_i\}_{i=1}^{|I|}) - \mathcal{R}$ as

$$
\mathcal{R}(\{\mathcal{D}_i\}_{i=1}^{|I|}) - \mathcal{R} =\mathbb{E}_{\mathcal{T}_i\sim\hat{p}(\mathcal{T})}\mathbb{E}_{(\mathbf{X}_i,\mathbf{Y}_i)\sim\hat{p}(\mathcal{T}_i)}\mathcal{L}(f_\theta(\mathbf{X}_i),\mathbf{Y}_i) - \mathbb{E}_{\mathcal{T}_i\sim p(\mathcal{T})}\mathbb{E}_{(\mathbf{X}_i,\mathbf{Y}_i)\sim\mathcal{T}_i}[\mathcal{L}(f_\theta(\mathbf{X}_i),\mathbf{Y}_i)]
$$

$$
= \underbrace{\mathbb{E}_{\mathcal{T}_i\sim\hat{p}(\mathcal{T})}\mathbb{E}_{(\mathbf{X}_i,\mathbf{Y}_i)\sim\hat{p}(\mathcal{T}_i)}\mathcal{L}(f_\theta(\mathbf{X}_i),\mathbf{Y}_i) - \mathbb{E}_{\mathcal{T}_i\sim\hat{p}(\mathcal{T})}\mathbb{E}_{(\mathbf{X}_i,\mathbf{Y}_i)\sim\mathcal{T}_i}[\mathcal{L}(f_\theta(\mathbf{X}_i),\mathbf{Y}_i)]}_{(i)}
$$

$$
+ \underbrace{\mathbb{E}_{\mathcal{T}_i\sim\hat{p}(\mathcal{T})}\mathbb{E}_{(\mathbf{X}_i,\mathbf{Y}_i)\sim\mathcal{T}_i}\mathcal{L}(f_\theta(\mathbf{X}_i),\mathbf{Y}_i) - \mathbb{E}_{\mathcal{T}_i\sim p(\mathcal{T})}\mathbb{E}_{(\mathbf{X}_i,\mathbf{Y}_i)\sim\mathcal{T}_i}[\mathcal{L}(f_\theta(\mathbf{X}_i),\mathbf{Y}_i)]}_{(ii)}.
$$

$$
\tag{16}
$$

Recall that we consider the function $f_\theta(\mathbf{x}) = \theta^{\top}\mathbf{x}$ and the function class

$$
\mathcal{W}_\gamma := \{\mathbf{x}\to\theta^{\top}\mathbf{x},\text{ such that }\theta\text{ satisfying }\mathbb{E}_{\mathbf{x}}\left[\psi(\langle\mathbf{x}-(\mathbf{c}_1+\mathbf{c}_2)/2,\theta\rangle)\right]\cdot\theta^{\top}\Sigma_X\theta\le\gamma\}, \tag{17}
$$

For each $\mathcal{T}_i$, let us consider $f_\theta(\cdot)\in\mathcal{W}_\gamma$. Combining Theorem 3.4 and Theorem A.1 in Zhang et al. (2021), we have the following result for the Rademacher complexity:

$$
R(\mathcal{F}_\mathcal{T};z_1,...,z_n) \le 2\max\{(\frac{\gamma}{\rho})^{1/4},(\frac{\gamma}{\rho})^{1/2}\}\sqrt{\frac{rank(\Sigma_X)}{N}}
$$

$$
\le 2\max\{(\frac{\gamma}{\rho})^{1/4},(\frac{\gamma}{\rho})^{1/2}\}\cdot\sqrt{\frac{r_\Sigma}{N}}.
$$

Then the first term (i) in Eqn. (16) can be bounded as below.

$$
\mathbb{E}_{\mathcal{T}_i\sim\hat{p}(\mathcal{T})}\mathbb{E}_{(\mathbf{X}_i,\mathbf{Y}_i)\sim\hat{p}(\mathcal{T}_i)}\mathcal{L}(f_\theta(\mathbf{X}_i),\mathbf{Y}_i) - \mathbb{E}_{\mathcal{T}_i\sim\hat{p}(\mathcal{T})}\mathbb{E}_{(\mathbf{X}_i,\mathbf{Y}_i)\sim\mathcal{T}_i}[\mathcal{L}(f_\theta(\mathbf{X}_i),\mathbf{Y}_i)]
$$

$$
\le\mathbb{E}_{T_i\sim\hat{p}(\mathcal{T})}|\mathbb{E}_{(\mathbf{X}_i,\mathbf{Y}_i)\sim\hat{p}(\mathcal{T}_i)}\mathcal{L}(f_\theta(\mathbf{X}_i),\mathbf{Y}_i) - \mathbb{E}_{(\mathbf{X}_i,\mathbf{Y}_i)\sim\mathcal{T}_i}[\mathcal{L}(f_\theta(\mathbf{X}_i),\mathbf{Y}_i)]
$$

$$
\le C_1\max\{(\frac{\gamma}{\rho})^{1/4},(\frac{\gamma}{\rho})^{1/2}\}\sqrt{\frac{r_\Sigma}{N}} + C_2\sqrt{\frac{\log(|I|/\delta)}{N}},
$$

where $C_1$ and $C_2$ are constants, and the additional $\log(|I|)$ term in the last inequality above since we take union bound on $|I|$ tasks.

Denote function $g:\mathcal{T}\to\mathbb{R}$ such that $g(\mathcal{T}) = \mathbb{E}_{(\mathbf{X},\mathbf{Y})\sim\mathcal{D}}(\mathcal{L}(f_\theta(\mathbf{X}),\mathbf{Y}))$. Denote

$$
\mathcal{G} = \{g(\mathcal{T}):g(\mathcal{T}) = \mathbb{E}_{(\mathbf{X},\mathbf{Y})\sim\mathcal{D}}(\mathcal{L}(f_\theta(\mathbf{X}),\mathbf{Y})),f_\theta\in\mathcal{W}_\gamma\}.
$$

Recall that $A(x) = 1/(1+e^x)$. The second term (ii) in Eqn. (16) requires computing the Rademacher complexity for the function class over distributions

$$\mathcal{R}(\mathcal{G}; \mathcal{T}_1, ..., \mathcal{T}_{|I|}) = \mathbb{E} \sup_{g \in \mathcal{G}} \frac{1}{|I|} |\sum_{i=1}^{|I|} \sigma_i g(\mathcal{T}_i)| = \mathbb{E} \sup_{g \in \mathcal{G}} \frac{1}{|I|} |\sum_{i=1}^{|I|} \sigma_i \mathbb{E}_{(\mathbf{X},\mathbf{Y})\sim\mathcal{T}_i}(A(\theta^\top \mathbf{X}) - \mathbf{X}\mathbf{Y}|$$

$$\lesssim \mathbb{E} \sup_{g \in \mathcal{G}} \frac{1}{|I|} |\sum_{i=1}^{|I|} \sigma_i \mathbb{E}_{(\mathbf{X},\mathbf{Y})\sim\mathcal{T}_i} |\theta^\top \mathbf{X}|| + \mathbb{E} \sup_{g \in \mathcal{G}} \frac{1}{|I|} |\sum_{i=1}^{|I|} \sigma_i \mathbb{E}_{(\mathbf{X},\mathbf{Y})\sim\mathcal{T}_i} \mathbf{Y}|$$

$$\lesssim \max\{(\frac{\gamma}{\rho})^{1/4}, (\frac{\gamma}{\rho})^{1/2}\} \sqrt{\frac{r_\Sigma}{|I|}} + \sqrt{\frac{1}{|I|}}.$$

Then we have the following bound on (ii) in Eqn. (16):

$$\mathbb{E}_{\mathcal{T}_i \sim \hat{p}(\mathcal{T})} \mathbb{E}_{(\mathbf{X}_i,\mathbf{Y}_i)\sim\mathcal{T}_i} \mathcal{L}(f_\theta(\mathbf{X}_i), \mathbf{Y}_i) - \mathbb{E}_{\mathcal{T}_i \sim p(\mathcal{T})} \mathbb{E}_{(\mathbf{X}_i,\mathbf{Y}_i)\sim\mathcal{T}_i} [\mathcal{L}(f_\theta(\mathbf{X}_i), \mathbf{Y}_i)]$$

$$\leq C_3 \max\{(\frac{\gamma}{\rho})^{1/4}, (\frac{\gamma}{\rho})^{1/2}\} \sqrt{\frac{r_\Sigma}{|I|}} + C_4 \sqrt{\frac{\log(1/\delta)}{|I|}}. \tag{18}$$

Combining the above pieces, we obtain the desired result. With probability at least $1 - \delta$,

$$|\mathcal{R}(\{\mathcal{D}_i\}_{i=1}^{|I|}) - \mathcal{R}| \leq 2B_1 \cdot \max\{(\frac{\gamma}{\rho})^{1/4}, (\frac{\gamma}{\rho})^{1/2}\} \cdot \left(\sqrt{\frac{r_\Sigma}{|I|}} + \sqrt{\frac{r_\Sigma}{N}}\right)$$

$$+ B_2 \sqrt{\frac{\log(1/\delta)}{2|I|}} + B_3 \sqrt{\frac{\log(|I|/\delta)}{N}}.$$

□

## B.2 THEORETICAL RESULTS UNDER THE LABEL-SHARING SCENARIO

As discussed in Line 131-133 of the main paper, for protonet, it is impractical to calculate the prototypes with mixed labels. Thus, under the label-sharing scenario, we only analyze the generalization ability of gradient-based meta-learning. Follow the assumptions under the non-label-sharing scenario, we first present the counterpart of Lemma 1 of the main paper.

**Lemma 4.** *Consider the MLTI with $\lambda \sim \text{Beta}(\alpha, \beta)$. Let $\psi(u) = e^u/(1+e^u)^2$ and $N_{i,r}$ denote the number of samples from the class $r$ in task $\mathcal{T}_i$. There exists a constant $c > 0$, such that the second-order approximation of $\mathcal{L}_t(\{\mathcal{D}_{i,cr}\}_{i=1}^{|I|})$ is given by*

$$\mathcal{L}_t(\bar{\lambda} \cdot \{\mathcal{D}_i\}_{i=1}^{|I|}) + c \frac{1}{N|I|} \sum_{i=1}^{|I|} \sum_{k=1}^{N} \psi(\mathbf{h}_{i,k}^{1\top}\phi_i) \cdot \phi_i^\top (\frac{1}{|I|} \sum_{i=1}^{|I|} \frac{1}{N|I|} \sum_{i=1}^{|I|} \sum_{k=1}^{N|I|} \mathbf{h}_{i,k}^1 \mathbf{h}_{i,k}^{1\top})\phi_i, \tag{19}$$

*Proof.* Under the label-sharing scenario, the interpolated dataset $\mathcal{D}_{i,cr}^q = (\tilde{\mathbf{H}}_{i,cr}^{q,1}, \tilde{\mathbf{Y}}_{i,cr}^q) := \{(\mathbf{h}_{i,k,cr}^1, \mathbf{y}_{i,k,cr})\}_{k=1}^N$ is constructed as

$$\mathbf{h}_{i,k,cr}^1 = \lambda \mathbf{h}_{i,k}^1 + (1-\lambda)\mathbf{h}_{j,k'}^1, \quad \mathbf{y}_{i,k,cr} = \lambda \mathbf{Y}_{i,k} + (1-\lambda)\mathbf{y}_{j,k'},$$

where $\lambda \sim \text{Beta}(\alpha, \beta), j \sim U([|I|])$.

By Lemma 3.1 in Zhang et al. (2021) (with proof on page 13), the data augmentation equals in distribution with the following augmentation

$$\mathbf{h}_{i,k,cr}^1 = \lambda \mathbf{h}_{i,k}^1 + (1-\lambda)\mathbf{h}_{j,k'}^1,$$

with $\lambda \sim \frac{\alpha}{\alpha+\beta}\text{Beta}(\alpha+1, \beta) + \frac{\alpha}{\alpha+\beta}\text{Beta}(\alpha+1, \beta), j \sim U([|I|])$.

Then we apply the same proof technique as the proof of Lemma 1 and obtain that the Taylor expansion of $\mathcal{L}_t(\{\mathcal{D}_{i,cr}\}_{i=1}^{|I|})$ up to the second-order equals to

$$\frac{1}{|I|}\sum_{i=1}^{|I|}\mathcal{L}(\bar{\lambda}\mathcal{D}_i) + c\frac{1}{|I|}\sum_{i=1}^{|I|}\frac{1}{N}\sum_{k=1}^{N}\psi(\mathbf{h}_{i,k}^{1\top}\phi_i)\phi_i^\top Cov(\mathbf{h}_{i,k,cr}^1 \mid \mathbf{h}_{i,k}^1)\phi_i$$

$$=\frac{1}{|I|}\sum_{i=1}^{|I|}\mathcal{L}(\bar{\lambda}\mathcal{D}_i) + c\frac{1}{|I|}\sum_{i=1}^{|I|}\frac{1}{N}\sum_{k=1}^{N}\psi(\mathbf{h}_{i,k}^{1\top}\phi_i)\cdot\phi_i^\top(\frac{1}{|I|}\sum_{i=1}^{|I|}\frac{1}{N|I|}\sum_{k=1}^{N}\mathbf{h}_{i,k}^1\mathbf{h}_{i,k}^{1\top})\phi_i$$

$$=\mathcal{L}_t(\bar{\lambda}\{\mathcal{D}_i\}_{i=1}^{|I|}) + c\frac{1}{N|I|}\sum_{i=1}^{|I|}\sum_{k=1}^{N}\psi(\mathbf{h}_{i,k}^{1\top}\phi_i)\cdot\phi_i^\top(\frac{1}{|I|}\sum_{i=1}^{|I|}\frac{1}{N|I|}\sum_{i=1}^{|I|}\sum_{k=1}^{N}\mathbf{h}_{i,k}^1\mathbf{h}_{i,k}^{1\top})\phi_i,$$

where $c = \mathbb{E}_{D_\lambda}[\frac{(1-\lambda)^2}{\lambda^2}]$ and $D_\lambda = \frac{\alpha}{\alpha+\beta}\text{Beta}(\alpha+1,\beta) + \frac{\alpha}{\alpha+\beta}\text{Beta}(\alpha+1,\beta)$. $\qquad\square$

Given Lemma 4, the population version of the regularization term can be defined in the same form of Eq. (14) and therefore the generalization theorem and its corresponding conclusions are the same as Theorem 1 and conclusions in the main paper.

Besides, in this work, the regression setting is only well-defined under the label-sharing scenario. The theoretical analysis under the label-sharing scenario (i.e., Lemma 4) in Section B.2 are not specific to the classification setting and still hold in the regression setting.

### B.3 DISCUSSION ABOUT THE VARIANCE OF MLTI

From the above analysis, we can see that the second order of regularization depends on $Cov(\mathbf{h}_{i,k,cr}^1 \mid \mathbf{h}_{i,k}^1)$ in Eqn. (1) (gradient-based meta-learning) or $Cov(\mathbf{x}_{i,k,cr} \mid \mathbf{x}_{i,k})$ in Eqn. (14) (metric-based meta-learning). Let $G$ denote the random variable which takes a uniform distribution on the indices of the tasks. By using the law of total variance, we have $Cov(\mathbf{h}_{i,k,cr}^1 \mid \mathbf{h}_{i,k}^1) = \mathbb{E}[Cov(\mathbf{h}_{i,k,cr}^1 \mid G, \mathbf{h}_{i,k}^1)] + Cov(\mathbb{E}[\mathbf{h}_{i,k,cr}^1 \mid G, \mathbf{h}_{i,k}^1]) \geq \mathbb{E}[Cov(\mathbf{h}_{i,k,cr}^1 \mid G, \mathbf{h}_{i,k}^1)]$, where the later is the covariance matrix induced by the individual task interpolation, i.e., $i = j$ in the interpolation process.

## C ADDITIONAL DISCUSSIONS BETWEEN MLTI AND INDIVIDUAL TASK AUGMENTATION

As shown in Figure 1, MLTI directly densifies task distributions by generating more tasks rather than apply augmentation strategies to each individual tasks. Compared with individual task augmentation (e.g., (Yao et al., 2021; Ni et al., 2021)), the reasons why MLTI leads to more dense task distributions are summarized under both label-sharing and non-label-sharing settings.

- **Label-sharing Setting.** MLTI densifies the task distribution by enabling cross-task interpolation. For example, in Pose prediction, we not only interpolate samples within each object, but cross-task interpolation significantly increases the number of tasks. Assume we have two objects (O1 and O2), individual task interpolation approaches (e.g., Meta-Maxup) only generate more samples in O1 or O2, where only one object information is covered. However, MLTI further allows generating tasks with both O1 and O2 information by interpolating data samples from O1 and O2.

- **Non-label-sharing Setting.** MLTI also leads to more dense task distribution under the non-label-sharing setting. For example, in 2-way classification with 3 training classes (C0, C1, C2), there are three original tasks, i.e., three classification pairs (C0, C1), (C0, C2), (C1, C2). Individual task interpolation increases the number of samples for each classification pair by enabling data from mix(C0, C1), mix(C0, C2), mix(C1, C2). However, it does not distinguish pairs like (mix(C0, C1), mix(C0, C2)), whereas MLTI does by allowing cross-tasks interpolation.

# D    ADDITIONAL EXPERIMENTAL SETUP AND RESULTS UNDER LABEL-SHARING SCENARIO

## D.1    DETAILED DESCRIPTIONS OF DATASETS AND EXPERIMENTAL SETUP

Under the label-sharing scenario, We detail the four datasets as well as their corresponding base models. All hyperparameters are listed in Table 5, which are selected by the cross-validation. Notice that all baselines use the same base models and interpolation-based methods (i.e., MetaMix, Meta-Maxup, MLTI) use the same interpolation strategies.

**RainbowMNIST (RMNIST).** Follow Finn et al. (2019), we create the RainbowMNIST dataset by changing the size (full/half), color (red/orange/yellow/green/blue/indigo/violet) and angle ($0°$, $90°$, $180°$, $270°$) of the original MNIST dataset. Specifically, we combine training and test set of original MNIST data and randomly select 5,600 samples for each class. We then split the combined dataset and create a series of subdatasets, where each subdataset corresponds to one combination of image transformations and has 1,000 samples, where each class has 100 samples. Each task in RainbowMNIST is randomly sampled from one subdataset. We use 16/6/10 subdatasets for meta-training/validation/testing and list their corresponding combinations of image transformations as follows:

Meta-training combinations:

```
(red, full, 90°), (indigo, full, 0°), (blue, full, 270°), (orange, half,
270°), (green, full, 90°), (green, full, 270°), (orange, full, 180°),
(red, full, 180°), (green, full, 0°), (orange, full, 0°), (violet, full,
270°), (orange, half, 90°), (violet, half, 180°), (orange, full, 90°),
(violet, full, 180°), (blue, full, 90°)
```

Meta-validation combinations:

```
(indigo, half, 270°), (blue, full, 0°), (yellow, half, 180°), (yellow,
half, 0°), (yellow, half, 90°), (violet, half, 0°)
```

Meta-testing combinations:

```
(yellow, full, 270°), (red, full, 0°), (blue, half, 270°), (blue, half,
0°), (blue, half, 180°), (red, half, 270°), (violet, full, 90°), (blue,
half, 90°), (green, half, 270°), (red, half, 90°)
```

To analyze the effect of task number, we sequentially add more combinations, which are listed as follows:

```
(indigo, half, 180°), (indigo, full, 180°), (violet, half, 90°), (green,
full, 180°), (indigo, half, 0°), (yellow, full, 90°), (indigo, 0, 90°),
(indigo, full, 270°), (yellow, full, 0°), (red, half, 180°), (green,
half, 0°), (violet, half, 270°), (yellow, half, 270°), (red, full, 270°),
(orange, half, 180°), (orange, half, 0°), (green, half, 180°), (indigo,
half, 90°), (blue, full, 180°), (violet, full, 0°), (yellow, full, 180°),
(orange, full, 270°), (red, half, 0°), (green, half, 90°)
```

In RainbowMNIST, we apply the standard convolutional neural network with four convolutional blocks as the base learner, where each block contains 32 output channels. For MAML, we apply the task adaptation process on both the last convolutional block and the classifier. We further use CutMix (Yun et al., 2019) for task interpolation.

**Pose prediction.** Follow Yin et al. (2020), pose prediction aims to to predict the pose of each object relative to its canonical orientation. We use the released dataset from Yin et al. (2020) to evaluate the performance of MLTI, where 50 and 15 objects are used for meta-training and meta-testing. Each category includes 100 gray-scale images, and the size of each image is $128 \times 128$.

As for the base model, we follow Yin et al. (2020) and define the base model with three fixed blocks and four adaptive blocks, where MAML only performs task-specific adaptation on the adapted blocks. Each fixed block contains one convolutional layer and one batch normalization layer, where the number of the output channels in the three convolutional layers are set as 32, 48, 64, respectively.

After the second fixed block, we add one max pooling layer, where both the kernel size and stride are set as 2. The output of the fixed blocks is fed into a fixed Linear layer and reshaped to $14 \times 14 \times 1$, which is further treated as the input of adapted blocks. Each adapted block includes one convolutional layer and one batch normalization layer, where the number of output channels of all convolutional layer is set as 64. ReLU function is used as the activation layer for all blocks in this experiment. Manifold Mixup (Verma et al., 2019) is used for feature interpolation. All baselines are rerun under the same environment.

**NCI.** We use the "NCI balanced" dataset released in (NCI, 2018), where 9 subdatasets are included (i.e., NCI 1, 33, 41, 47, 81, 83, 109, 123, 145). Each NCI subdataset is a complete bioassay for an binary anticancer activity classification (i.e., positive/negative), where each assay contains a set of chemical compounds. We randomly sample 1000 data samples for each subdataset. In our experiments, we represent each drug compound through the 1024 bit fingerprint features extracted by RDKit (Landrum, 2016), where each fingerprint bit corresponds to a fragment of the molecule. We select NCI 41, 47, 81, 83, 109, 145 for meta-training and NCI 1, 33, 123 for meta-testing, where each task is sampled from one subdataset.

The extracted 1024 bit fingerprint features are fed into an neural network with two fully connected blocks and one linear regressor. Each fully connected block contains one linear layer, one batch normalization layer and one Leakyrelu function (negative slope: 0.01) as activation layer, where the number of output neurons of each fully connected block is set as 500. In our experiments, for MAML, the parameters in the first fully connected block is globally shared across all tasks, and the rest layers are set as adapted layers. We adopt Manifold Mixup (Verma et al., 2019) as the interpolation strategy.

**TDC Metabolism.** Similar to NCI dataset, we create another bio-related dataset – TDC Metabolism. In TDC Metabolism, we select 8 subdatasets related to drug metabolism from the whole TDC dataset (Huang et al., 2021), including CYP P450 2C19/2D6/3A4/1A2/2C9 Inhibition, CYP2C9/CYP2D6/CYP3A4 Substrate. The aim of each dataset is to predict whether each drug compound has the corresponding property. We use P450 1A2/3A4/2D6 and CYP2C9/CYP2D6 substrate for meta-training, and CYP2C19/2C9 and CYP3A4 substrate for meta-testing. We balance each subdataset by randomly selecting at most 1000 data samples and each task is randomly sampled from one subdataset. Analogy to the NCI dataset, we use the same neural network architecture and features (i.e., 1024 bit fingerprint) for TDC Metabolism. Manifold Mixup (Verma et al., 2019) is used as the interpolation strategy.

Table 5: Hyperparameters under the label-sharing scenario.

| Hyperparameters (MAML) | Pose | RMNIST | NCI | Metabolism |
|---|---|---|---|---|
| inner-loop learning rate | 0.01 | 0.01 | 0.01 | 0.01 |
| outer-loop learning rate | 0.001 | 0.001 | 0.001 | 0.001 |
| Beta$(\alpha, \beta)$, $\alpha = \beta$ | 0.5 ($i = j$), 0.1 ($i \neq j$) | 2.0 | 2.0 | 0.5 |
| num updates | 5 | 5 | 5 | 5 |
| batch size | 10 | 4 | 4 | 4 |
| query size for meta-training | 15 | 1 | 10 | 10 |
| maximum training iterations | 10,000 | 30,000 | 10,000 | 10,000 |

| Hyperparameters (ProtoNet) | Pose | RMNIST | NCI | Metabolism |
|---|---|---|---|---|
| learning rate | n/a | 0.001 | 0.001 | 0.001 |
| Beta$(\alpha, \beta)$, $\alpha = \beta$ | n/a | 2.0 | 0.5 | 0.5 |
| batch size | n/a | 4 | 4 | 4 |
| query size for meta-training | n/a | 1 | 10 | 10 |
| maximum training iterations | n/a | 30,000 | 10,000 | 10,000 |

## D.2 COMPATIBILITY ANALYSIS UNDER LABEL-SHARING SCENARIO

In Table 6, we show the additional compatibility analysis under the label-sharing scenario. We observe that MLTI achieves the best performance under different backbone meta-learning algorithms, indicating the compatibility and effectiveness of MLTI in improving the generalization ability.

Table 6: Additional compatibility analysis under the label-sharing scenario (evaluation metric: MSE for Pose and accuracy for other datasets), where the 95% confidence intervals are also reported.

| Model | | Pose (15-shot) | RMNIST (1-shot) | NCI (5-shot) | Metabolism (5-shot) |
|---|---|---|---|---|---|
| MatchingNet | | n/a | $73.87 \pm 1.24\%$ | $75.03 \pm 0.89\%$ | $60.95 \pm 0.94\%$ |
| | +MLTI | n/a | $\mathbf{75.36 \pm 0.81\%}$ | $\mathbf{76.81 \pm 0.77\%}$ | $\mathbf{63.02 \pm 1.09\%}$ |
| MetaSGD | | $2.227 \pm 0.098$ | $66.68 \pm 1.28\%$ | $77.74 \pm 0.82\%$ | $57.54 \pm 1.03\%$ |
| | +MLTI | $\mathbf{1.938 \pm 0.078}$ | $\mathbf{72.78 \pm 1.06\%}$ | $\mathbf{78.43 \pm 0.86\%}$ | $\mathbf{61.83 \pm 0.99\%}$ |
| ANIL | | $6.947 \pm 0.159$ | $56.52 \pm 1.18\%$ | $77.65 \pm 0.79\%$ | $57.63 \pm 1.07\%$ |
| | +MLTI | $\mathbf{6.042 \pm 0.146}$ | $\mathbf{64.63 \pm 1.47\%}$ | $\mathbf{78.46 \pm 0.75\%}$ | $\mathbf{60.34 \pm 1.01\%}$ |
| MC | | $2.174 \pm 0.096$ | $58.03 \pm 1.24\%$ | $77.25 \pm 0.80\%$ | $58.37 \pm 1.02\%$ |
| | +MLTI | $\mathbf{1.904 \pm 0.073}$ | $\mathbf{63.25 \pm 1.36\%}$ | $\mathbf{78.52 \pm 0.86\%}$ | $\mathbf{60.59 \pm 1.05\%}$ |

### D.3 EFFECT OF THE NUMBER OF META-TRAINING TASKS UNDER LABEL-SHARING SCENARIO

For RainbowMNIST, we analyze the effect of the number of meta-training combinations with respect to the performance, where the number of meta-training combinations directly reflects the number of meta-training tasks. Figure 4a and 4b illustrate the results of MAML and ProtoNet, respectively. The results indicate that MLTI consistently improves the performance, especially when the number of combinations are limited (e.g., Figure 4b).

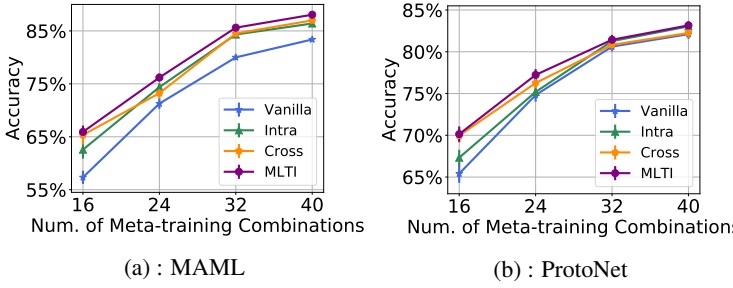

(a) : MAML  (b) : ProtoNet

Figure 4: Accuracy w.r.t. the number of meta-training combinations of transformations in RainbowMNIST. Intra and Cross represent the intra-task interpolation (i.e., $\mathcal{T}_i = \mathcal{T}_j$) and the cross-task interpolation (i.e., $\mathcal{T}_i \neq \mathcal{T}_j$), respectively.

## E ADDITIONAL EXPERIMENTAL SETUP AND RESULTS UNDER NON-LABEL-SHARING SCENARIO

### E.1 DETAILED DATASET DESCRIPTIONS OF EXPERIMENTAL SETUP

In this section, we detail the dataset description and the model architecture under the non-label-sharing scenario. The hyperparameters are selected by cross-validation and listed in Table 7. For fair comparison, all baselines adopt the same base models. Additionally, all interpolation-based methods (i.e., MetaMix, Meta-Maxup, MLTI) adopt the same interpolation strategies.

**miniImagenet-S.** In miniImagenet-S, we reduce the number of tasks by controlling the number of meta-training classes. Specifically, in miniImagenet-S, the following classes are used for meta-training:

```
n03017168, n07697537, n02108915, n02113712, n02120079, n04509417,
n02089867, n03888605, n04258138, n03347037, n02606052, n06794110
```

To analyze the effect of task number, we incrementally add more classes by the following sequence:

```
n03476684, n02966193, n13133613, n03337140, n03220513, n03908618,
n01532829, n04067472, n02074367, n03400231, n02108089, n01910747,
n02747177, n02795169, n04389033, n04435653, n02111277, n02108551,
```

```
n04443257, n02101006, n02823428, n03047690, n04275548, n04604644,
n02091831, n01843383, n02165456, n03676483, n04243546, n03527444,
n01770081, n02687172, n09246464, n03998194, n02105505, n01749939,
n04251144, n07584110, n07747607, n04612504, n01558993, n03062245,
n04296562, n04596742, n03838899, n02457408, n13054560, n03924679,
n03854065, n01704323, n04515003, n03207743
```

We apply the same base learner as Finn et al. (2017) in our experiments, which contains four convolutional blocks and a classifier layer. Each convolutional block includes a convolutional layer, a batch normalization layer and a ReLU activation layer. For MAML, we apply the task-specific adaptation on the last convolutional block and the classifier layer, which yields the best empirical performance.

**ISIC.** In ISIC dataset, we select task 3 in "ISIC 2018: Skin Lesion Analysis Towards Melanoma Detection" challenge (Milton, 2019), where 10,015 medical images are labeled by seven lesion categories: Nevus, Dermatofibroma, Melanoma, Pigmented Bowen's, Benign Keratoses, Basal Cell Carcinoma, Vascular. Follow Li et al. (2020), we use four categories with the largest number of categories as meta-training classes, including Nevus, Melanoma, Benign Keratoses, Basal Cell Carcinoma. The rest three categories are treated as meta-testing classes. We apply N-way, K-shot settings in ISIC and set $N = 2$ in our experiments. Thus, there are only six class combinations for the meta-training process. Each medical image in ISIC are re-scaled to the size of $84 \times 84 \times 3$ and the base model as well as other settings are the same as miniImagenet-S.

**DermNet-S.** We construct the Dermnet-S dataset from the public Dermnet Skin Disease Atlas (Der, 2016), which includes more than 22,000 across 625 fine-grained classes after removing duplicated images/classes. Similar to (Prabhu et al., 2018), we focus on the classes with no less than 30 images, resulting in 203 selected classes. The base model and other settings are the same as miniImagenet-S and ISIC. The selected classes has a long-tail and we use the top-30 classes for meta-training and the bottom-53 classes for meta-testing. The detailed meta-training and meta-testing classes are listed as follows.

Meta-training classes:
```
Seborrheic Keratoses Ruff, Herpes Zoster, Atopic Dermatitis Adult
Phase, Psoriasis Chronic Plaque, Eczema Hand, Seborrheic Dermatitis,
Keratoacanthoma, Lichen Planus, Epidermal Cyst, Eczema Nummular, Tinea
(Ringworm) Versicolor, Tinea (Ringworm) Body, Lichen Simplex Chronicus,
Scabies, Psoriasis Palms Soles, Malignant Melanoma, Candidiasis large
Skin Folds, Pityriasis Rosea, Granuloma Annulare, Erythema Multiforme,
Seborrheic Keratosis Irritated, Stasis Dermatitis and Ulcers, Distal
Subungual Onychomycosis, Allergic Contact Dermatitis, Psoriasis,
Molluscum Contagiosum, Acne Cystic, Perioral Dermatitis, Vasculitis,
Eczema Fingertips
```

Meta-testing classes:
```
Warts, Ichthyosis Sex Linked, Atypical Nevi, Venous Lake, Erythema
Nodosum, Granulation Tissue, Basal Cell Carcinoma Face, Acne Closed
Comedo, Scleroderma, Crest Syndrome, Ichthyosis Other Forms, Psoriasis
Inversus, Kaposi Sarcoma, Trauma, Polymorphous Light Eruption,
Dermagraphism, Lichen Sclerosis Vulva, Pseudomonas, Cutaneous Larva
Migrans, Psoriasis Nails, Corns, Lichen Sclerosus Penis, Staphylococcal
Folliculitis, Chilblains Perniosis, Psoriasis Erythrodermic, Squamous
Cell Carcinoma Ear, Basal Cell Carcinoma Ear, Ichthyosis Dominant,
Erythema Infectiosum, Actinic Keratosis Hand, Basal Cell Carcinoma Lid,
Amyloidosis, Spiders, Erosio Interdigitalis Blastomycetica, Scarlet
Fever, Pompholyx, Melasma, Eczema Trunk Generalized, Metastasis, Warts
Cryotherapy, Nevus Spilus, Basal Cell Carcinoma Lip, Enterovirus,
Pseudomonas Cellulitis, Benign Familial Chronic Pemphigus, Pressure
Urticaria, Halo Nevus, Pityriasis Alba, Pemphigus Foliaceous, Cherry
Angioma, Chapped Fissured Feet, Herpes Buttocks, Ridging Beading
```

To further analyze the effect of task number, similar to miniImagenet, we incrementally add more classes for meta-training by the following sequence:

```
Lupus Chronic Cutaneous, Rosacea, Genital Warts, Dermatofibroma,
Seborrheic Keratoses Smooth, Basal Cell Carcinoma Lesion, Sun Damaged
Skin, Tinea (Ringworm) Groin, Lichen Sclerosus Skin, Atopic Dermatitis
Childhood Phase, Psoriasis Guttate, Warts Common, Warts Plantar,
Herpes Cutaneous, Eczema Subacute, Psoriasis Scalp, Bullous Pemphigoid,
Sebaceous Hyperplasia, Pyogenic Granuloma, Phototoxic Reactions,
Urticaria Acute, CTCL Cutaneous T-Cell Lymphoma, Drug Eruptions,
Mucous Cyst, Alopecia Areata, Hidradenitis Suppurativa, Herpes Type
1 Recurrent, Viral Exanthems, Skin Tags Polyps, Melanocytic Nevi,
Dermatitis Herpetiformis, Eczema Foot, Morphea, Intertrigo, Atopic
Dermatitis Infant phase, Bowen Disease, Necrobiosis Lipoidica, Lentigo
Adults, Xanthomas, Rhus Dermatitis, Keratosis Pilaris, Schamberg Disease,
Rosacea Nose, Chondrodermatitis Nodularis, Keloids, Tinea (Ringworm) Foot
Webs, Tinea (Ringworm) Laboratory, Porokeratosis, Impetigo, Basal Cell
Carcinoma Pigmented, Porphyrias, Epidermal Nevus, Fixed Drug Eruption,
Venous Malformations, Acne Open Comedo, Perlèche, Acne Pustular, Herpes
Type 1 Primary, Tinea (Ringworm) Scalp, Neurofibromatosis, Warts Flat,
Pityriasis Rubra Pilaris, Hemangioma, Herpes Type 2 Primary, Tinea
(Ringworm) Hand Dorsum, Neurotic Excoriations, Tinea (Ringworm) Primary
Lesion, Basal Cell Carcinoma Nose, Dariers disease, Tinea (Ringworm) Foot
Dorsum, Tinea (Ringworm) Face, Tinea (Ringworm) Incognito, Acanthosis
Nigricans, Onycholysis, Warts Digitate, Psoriasis Pustular Generalized,
Varicella, Basal Cell Carcinoma Superficial, Herpes Simplex, Nevus
Sebaceous, Actinic Keratosis 5 FU, Acne Keloidalis, Hemangioma Infancy,
Candida Penis, Tuberous Sclerosis, Stucco Keratoses, Eczema Herpeticum,
Dyshidrosis, Epidermolysis Bullosa, Actinic Cheilitis Squamous Cell
Lip, Ticks, Actinic Keratosis Face, Chronic Paronychia, Biting Insects,
Dermatomyositis, Grovers Disease, Atypical Nevi Dermoscopy, Patch
Testing, Telangiectasias, Pityriasis Lichenoides, Psoriasis Hand, Actinic
Keratosis Lesion, Lichen Planus Oral, Tinea (Ringworm) Foot Plantar,
Eczema Chronic, Herpes Type 2 Recurrent, Lupus Acute, Eczema Asteatotic,
Pilar Cyst, Pemphigus, Vitiligo, Keratolysis Exfoliativa, AIDS (Acquired
Immunodeficiency Syndrome), Syringoma, Habit Tic Deformity, Congenital
Nevus, Angiokeratomas, Prurigo Nodularis, Pediculosis Pubic, Tinea
(Ringworm) Palm
```

We use CutMix (Yun et al., 2019) to interpolate samples in the above three image classification datasets. Besides, the interpolation strategy is applied on the query set when $i = j$, which empirically achieves better performance.

**Tabular Murris.** Follow (Cao et al., 2021), the Tabular Murris dataset is collected from 23 organs, which contains 105,960 cells of 124 cell types. We aim to classify the cell type of each cell, which is represented by 2,866 genes (i.e, the dimension of features is 2,866). We use the code of Cao et al. (2021) to construct tasks, where 15/4/4 organs are selected for meta-training/validation/testing. The selected organs are detailed as follows:

Meta-training organs:
```
BAT, MAT, Limb Muscle, Trachea, Heart, Spleen, GAT, SCAT, Mammary Gland,
Liver, Kidney, Bladder, Brain Myeloid, Brain Non-Myeloid, Diaphragm.
```

Meta-validation organs:
```
Skin, Lung, Thymus, Aorta
```

Meta-testing organs:
```
Large Intestine, Marrow, Pancreas, Tongue
```

In Tabular Murris, the base model contains two fully connected blocks and a linear regressor, where each fully connected block contains a linear layer, a batch normalization layer, a ReLU activation layer, and a dropout layer. Follow Cao et al. (2021), the default dropout ratio and the output channels

of the linear layer are set as 0.2, 64, respectively. We apply Mainfold Mixup (Verma et al., 2019) as the interpolation strategy. It also worthwhile to mention that the performance of gradient-based methods (e.g., MAML) significantly outperforms the reported results in Cao et al. (2021) since they only apply 1-step inner-loop gradient descent in their released code. In addition, during the whole meta-testing process, we change the mode from training to evaluation, resulting in the better performance of metric-based methods (e.g., Protonet).

Table 7: Hyperparameters under the non-label-sharing scenario.

| Hyperparameters (MAML) | miniImagenet-S | ISIC | DermNet-S | Tabular Murris |
|---|---|---|---|---|
| inner-loop learning rate | 0.01 | 0.01 | 0.01 | 0.01 |
| outer-loop learning rate | 0.001 | 0.001 | 0.001 | 0.001 |
| Beta($\alpha, \beta$), $\alpha = \beta$ | 2.0 | 2.0 | 2.0 | 2.0 |
| num updates | 5 | 5 | 5 | 5 |
| batch size | 4 | 4 | 4 | 4 |
| query size for meta-training | 15 | 15 | 15 | 15 |
| maximum training iterations | 50,000 | 50,000 | 50,000 | 10,000 |

| Hyperparameters (ProtoNet) | Pose | RMNIST | NCI | Metabolism |
|---|---|---|---|---|
| learning rate | 0.001 | 0.001 | 0.001 | 0.001 |
| Beta($\alpha, \beta$), $\alpha = \beta$ | 2.0 | 2.0 | 0.5 | 0.5 |
| batch size | 4 | 4 | 4 | 4 |
| query size for meta-training | 15 | 15 | 15 | 15 |
| maximum training iterations | 50,000 | 50,000 | 50,000 | 10,000 |

## E.2 RESULTS ON FULL-SIZE FEW-SHOT IMAGE CLASSIFICATION DATASETS

In this subsection, we provide the results of MLTI and other strategies on full-size miniImagenet and DermNet in Table 8, where 64 and 150 training classes are used in the meta-training process, respectively. Under the full-size miniImagenet and DermNet settings, the original meta-training tasks are sufficient to obtain satisfying performance. Nevertheless, applying MLTI also outperforms other strategies, demonstrating its effectiveness in improving generalization ability in meta-learning.

## E.3 COMPATIBILITY ANALYSIS UNDER NON-LABEL-SHARING SCENARIO

In Table 9, we report the results of additional compatibility analysis under the non-label-sharing scenario. The results validate the effectiveness and compatibility of the proposed MLTI.

## E.4 RESULTS OF ABLATION STUDY UNDER NON-LABEL-SHARING SCENARIO

In Table 10, we report the ablation study under the non-label-sharing scenario. The results indicate that MLTI outperforms all other ablation strategies and achieves better generalization ability.

## E.5 ADDITIONAL ANALYSIS ABOUT MODEL CAPACITY AND HYPERPARAMETERS

### E.5.1 MODEL CAPACITY ANALYSIS

Here, we investigate the performance of MLTI with a heavier backbone model. To increase the model capacity, we use ResNet-12 as the base model. The results on miniImagenet-S and Dermnet-S are reported in Table 11. Here, the results of Meta-Maxup and MetaMix are also reported for comparison. According to the results, MLTI outperforms vanilla MAML/ProtoNet, Meta-Maxup and MetaMix, verifying its effectiveness even with a larger base model.

### E.5.2 ANALYSIS OF THE INTERPOLATION LAYERS

We further conduct experiments on Metabolism and Tabular Murris to analyze the performance with different interpolation layers when Manifold Mixup (i.e., interpolating features) is used for data interpolation. Here, ProtoNet is used as backbone. We report the results in Table 12. The results indicate (1) fixing the interpolation layer can also boost the performance; (2) randomly selecting

Table 8: Results (averaged accuracy ± 95% confidence interval) of full-size miniImagenet and DermNet.

| Backbone | Strategies | miniImagenet-full | | DermNet-full | |
|---|---|---|---|---|---|
| | | 1-shot | 5-shot | 1-shot | 5-shot |
| MAML | Vanilla | 46.90 ± 0.79% | 63.02 ± 0.68% | 49.58 ± 0.83% | 69.15 ± 0.69% |
| | Meta-Reg | 47.02 ± 0.77% | 63.19 ± 0.69% | 50.10 ± 0.86% | 69.73 ± 0.70% |
| | TAML | 46.40 ± 0.82% | 63.26 ± 0.68% | 50.26 ± 0.85% | 69.40 ± 0.75% |
| | Meta-Dropout | 47.47 ± 0.81% | 64.11 ± 0.71% | 51.10 ± 0.84% | 69.08 ± 0.69% |
| | MetaMix | 47.81 ± 0.78% | 64.22 ± 0.68% | 51.83 ± 0.83% | 71.57 ± 0.67% |
| | Meta-Maxup | 47.68 ± 0.79% | 63.51 ± 0.75% | 51.95 ± 0.88% | 70.84 ± 0.68% |
| | **MLTI (ours)** | **48.62 ± 0.76%** | **64.65 ± 0.70%** | **52.32 ± 0.88%** | **71.77 ± 0.67%** |
| ProtoNet | Vanilla | 47.05 ± 0.79% | 64.03 ± 0.68% | 49.91 ± 0.79% | 67.45 ± 0.70% |
| | MetaMix | 47.21 ± 0.76% | 64.38 ± 0.67% | 51.50 ± 0.76% | 69.55 ± 0.68% |
| | Meta-Maxup | 47.33 ± 0.79% | 64.43 ± 0.69% | 51.18 ± 0.83% | 69.07 ± 0.72% |
| | **MLTI (ours)** | **48.11 ± 0.81%** | **65.22 ± 0.70%** | **52.91 ± 0.81%** | **71.30 ± 0.69%** |

Table 9: Additional compatibility analysis under the setting of the non-label-sharing scenario. We show averaged accuracy ± 95% confidence interval.

| | Model | | miniImagenet-S | ISIC | DermNet-S | Tabular Muris |
|---|---|---|---|---|---|---|
| 1-shot | MatchingNet | | 39.40 ± 0.70% | 61.01 ± 1.00% | 46.50 ± 0.84% | 80.37 ± 0.90% |
| | | +MLTI | **42.09 ± 0.81%** | **63.87 ± 1.08%** | **49.11 ± 0.86%** | **81.72 ± 0.89%** |
| | MetaSGD | | 37.98 ± 0.75% | 58.03 ± 0.79% | 41.56 ± 0.80% | 81.55 ± 0.91% |
| | | +MLTI | **39.58 ± 0.76%** | **61.57 ± 1.10%** | **45.49 ± 0.83%** | **83.31 ± 0.87%** |
| | ANIL | | 37.66 ± 0.77% | 59.08 ± 1.04% | 43.88 ± 0.82% | 75.67 ± 0.99% |
| | | +MLTI | **39.15 ± 0.73%** | **61.78 ± 1.24%** | **46.79 ± 0.77%** | **77.11 ± 1.00%** |
| | MC | | 37.43 ± 0.75% | 58.77 ± 1.06% | 43.09 ± 0.86% | 80.47 ± 0.91% |
| | | +MLTI | **40.22 ± 0.77%** | **61.53 ± 0.79%** | **47.40 ± 0.83%** | **82.44 ± 0.88%** |
| 5-shot | MatchingNet | | 50.21 ± 0.68% | 70.16 ± 0.72% | 62.56 ± 0.71% | 85.99 ± 0.76% |
| | | +MLTI | **54.59 ± 0.72%** | **73.62 ± 0.84%** | **65.65 ± 0.71%** | **87.75 ± 0.60%** |
| | MetaSGD | | 49.52 ± 0.73% | 68.01 ± 0.87% | 58.97 ± 0.73% | 91.03 ± 0.55% |
| | | +MLTI | **53.19 ± 0.69%** | **70.44 ± 0.65%** | **63.86 ± 0.71%** | **92.05 ± 0.51%** |
| | ANIL | | 49.21 ± 0.70% | 69.48 ± 0.66% | 60.54 ± 0.76% | 81.32 ± 0.89% |
| | | +MLTI | **52.76 ± 0.72%** | **72.01 ± 0.68%** | **63.07 ± 0.71%** | **82.75 ± 0.89%** |
| | MC | | 49.66 ± 0.69% | 68.29 ± 0.85% | 60.03 ± 0.72% | 89.30 ± 0.56% |
| | | +MLTI | **53.42 ± 0.71%** | **70.58 ± 0.82%** | **63.10 ± 0.68%** | **91.23 ± 0.52%** |

the interpolation layer achieves the best performance; (3) interpolating at the lower layer performs similarly as interpolating at the higher layer, indicating the robustness of MLTI with different selected layers.

### E.6 ADDITIONAL RESULTS OF ANALYSIS ABOUT THE NUMBER OF TASKS

Besides the results in the main paper, we further provide the 1-shot results for miniImagenet and Dermnet in Figure 5a and 5b, respectively. The results corroborate our findings in the main paper that MLTI consistently improves the performance, especially when the number of tasks is limited.

### E.7 MLTI WITH EXTREMELY LIMITED TASKS

In this section, we investigate how MLTI performs when we only have extremely limited tasks. Here, we decrease the number of distinct meta-training tasks of miniImagenet and DermNet to 56 by reducing the number of base classes to 8 since $\binom{8}{5} = 56$. Under this setting, two additional baselines with supervised training process (SL) (Dhillon et al., 2020) and multi-task training process

Table 10: Ablation study under the non-label-sharing scenario. We find that MLTI performs best.

| Backbone | Strategies | miniImagenet-S | ISIC | DermNet-S | Tabular Murris |
|---|---|---|---|---|---|
| MAML (1-shot) | Vanilla | $38.27 \pm 0.74\%$ | $57.59 \pm 0.79\%$ | $43.47 \pm 0.83\%$ | $79.08 \pm 0.91\%$ |
| | Intra-Intrpl | $39.31 \pm 0.75\%$ | $60.39 \pm 0.93\%$ | $47.16 \pm 0.86\%$ | $81.49 \pm 0.91\%$ |
| | Cross-Intrpl | $39.91 \pm 0.74\%$ | $61.06 \pm 1.23\%$ | $46.21 \pm 0.79\%$ | $80.65 \pm 0.92\%$ |
| | **MLTI (ours)** | $\mathbf{41.58 \pm 0.72\%}$ | $\mathbf{61.79 \pm 1.00\%}$ | $\mathbf{48.03 \pm 0.79\%}$ | $\mathbf{81.73 \pm 0.89\%}$ |
| MAML (5-shot) | Vanilla | $52.14 \pm 0.65\%$ | $65.24 \pm 0.77\%$ | $60.56 \pm 0.74\%$ | $88.55 \pm 0.60\%$ |
| | Intra-Intrpl | $52.74 \pm 0.74\%$ | $68.96 \pm 0.74\%$ | $63.65 \pm 0.70\%$ | $89.89 \pm 0.62\%$ |
| | Cross-Intrpl | $53.34 \pm 0.77\%$ | $70.20 \pm 0.70\%$ | $62.59 \pm 0.76\%$ | $89.97 \pm 0.56\%$ |
| | **MLTI (ours)** | $\mathbf{55.22 \pm 0.76\%}$ | $\mathbf{70.69 \pm 0.68\%}$ | $\mathbf{64.55 \pm 0.74\%}$ | $\mathbf{91.08 \pm 0.54\%}$ |
| ProtoNet (1-shot) | Vanilla | $36.26 \pm 0.70\%$ | $58.56 \pm 1.01\%$ | $44.21 \pm 0.75\%$ | $80.03 \pm 0.90\%$ |
| | Intra-Intrpl | $39.31 \pm 0.75\%$ | $60.70 \pm 1.16\%$ | $46.97 \pm 0.81\%$ | $80.56 \pm 0.94\%$ |
| | Cross-Intrpl | $40.95 \pm 0.76\%$ | $62.22 \pm 1.19\%$ | $48.68 \pm 0.85\%$ | $81.22 \pm 0.90\%$ |
| | **MLTI (ours)** | $\mathbf{41.36 \pm 0.75\%}$ | $\mathbf{62.82 \pm 1.13\%}$ | $\mathbf{49.38 \pm 0.85\%}$ | $\mathbf{81.89 \pm 0.88\%}$ |
| ProtoNet (5-shot) | Vanilla | $50.72 \pm 0.70\%$ | $66.25 \pm 0.96\%$ | $60.33 \pm 0.70\%$ | $89.20 \pm 0.56\%$ |
| | Intra-Intrpl | $53.33 \pm 0.68\%$ | $70.12 \pm 0.88\%$ | $62.91 \pm 0.75\%$ | $89.78 \pm 0.58\%$ |
| | Cross-Intrpl | $54.62 \pm 0.72\%$ | $71.47 \pm 0.89\%$ | $64.32 \pm 0.71\%$ | $90.05 \pm 0.57\%$ |
| | **MLTI (ours)** | $\mathbf{55.34 \pm 0.74\%}$ | $\mathbf{71.52 \pm 0.89\%}$ | $\mathbf{65.19 \pm 0.73\%}$ | $\mathbf{90.12 \pm 0.59\%}$ |

Table 11: Analysis on the heavier base model (ResNet-12) under 1-shot miniImagenet-S and DermNet-S settings.

| Backbone | Strategies | miniImagenet-S | DermNet-S |
|---|---|---|---|
| MAML | Vanilla | $40.02 \pm 0.78\%$ | $47.58 \pm 0.93\%$ |
| | MetaMix | $42.26 \pm 0.75\%$ | $51.40 \pm 0.89\%$ |
| | Meta-Maxup | $41.97 \pm 0.78\%$ | $50.82 \pm 0.85\%$ |
| | **MLTI (ours)** | $\mathbf{43.35 \pm 0.80\%}$ | $\mathbf{52.03 \pm 0.90\%}$ |
| ProtoNet | Vanilla | $40.96 \pm 0.75\%$ | $48.65 \pm 0.85\%$ |
| | MetaMix | $42.95 \pm 0.87\%$ | $51.18 \pm 0.90\%$ |
| | Meta-Maxup | $42.68 \pm 0.78\%$ | $50.96 \pm 0.88\%$ |
| | **MLTI (ours)** | $\mathbf{44.08 \pm 0.83\%}$ | $\mathbf{52.01 \pm 0.93\%}$ |

(MTL) (Wang et al., 2021) are also used for comparison. We also report the results of the best baseline – MetaMix. All results are listed in Table 13 and corroborate the effectiveness of MLTI even with extremely limited meta-training tasks.

### E.8 RESULTS ON ADDITIONAL DATASETS

We further provided two additional datasets under the non-label-sharing setting to show the effectiveness of MLTI – tieredImageNet-S and Huffpost. Both datasets are non-label-sharing datasets. We detail the data descriptions and hyperparameters in the following.

- **tieredImageNet-S**. tieredImageNet (Ren et al., 2018) is a few-shot image classification dataset, which consists of 351/97/160 images for meta-training/validation/testing. Following miniImagenet-S and DermNet-S, we use 35 original meta-training classes in tieredImageNet. All hyperparameters, base model and interpolation strategies are set as the same as miniImagenet-S.

- **Huffpost**. Huffpost (Misra, 2018) aims to classify the category for each sentence. We follow Bao et al. (2020) to preprocess Huffpost data, where 25/6/10 classes are used for meta-training/validation/testing. In our experiments, to construct the base model, we use ALBERT (Lan et al., 2019) as the encoder and use two fully connected layers as the classifier. The query set size for training and testing is set as 5. Due to the memory limitation, we set the batch size as 1 and the learning rate (outer-loop learning rate) for MAML as 2e-5. The inner-loop learning rate for

Table 12: Analysis of interpolation layers. Layer 0, 1 represents randomly select layer 0 or layer 1 for interpolation. None means vanilla ProtoNet.

| Interpolation Layer | Metabolism: 5-shot | Tabular Murris: 1-shot |
|---|---|---|
| None | $61.06 \pm 0.94\%$ | $80.03 \pm 0.90\%$ |
| Layer 0 | $62.53 \pm 0.98\%$ | $81.18 \pm 0.93\%$ |
| Layer 1 | $62.38 \pm 0.94\%$ | $81.25 \pm 0.90\%$ |
| Layer 0, 1 | $\mathbf{63.47 \pm 0.96\%}$ | $\mathbf{81.89 \pm 0.88\%}$ |

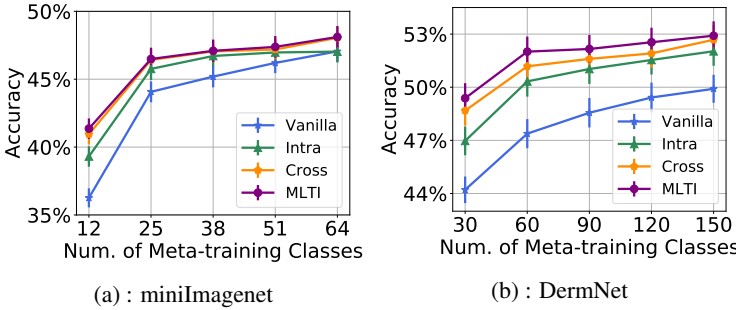

(a) : miniImagenet

(b) : DermNet

Figure 5: Accuracy w.r.t. the number of meta-training classes under the non-label-sharing scenario (1-shot). Intra and Cross represent the intra-task interpolation (i.e., $\mathcal{T}_i = \mathcal{T}_j$) and the cross-task interpolation (i.e., $\mathcal{T}_i \neq \mathcal{T}_j$), respectively.

MAML is set as 0.01. We set $\alpha = \beta = 2.0$ in $\mathrm{Beta}(\alpha, \beta)$. The number of inner loop updates in MAML is set as 5 and the maximum training iteration is set as 10,000. Manifold Mixup is used for data interpolation.

We report the results in Table 14. In these two additional datasets, MLTI also outperforms other methods, showing its promise in improving generalization in meta-learning.

### E.9 Full Tables with Confidence Interval

Table 15, 16 report the full results (accuracy $\pm$ 95% confidence interval) of Table 3, 4 in the paper.

Table 13: Results of MLTI with extremely limited tasks. SL and MTL represent methods with supervised and multi-task training process, respectively.

| Model | | miniImagenet-S (8 classes) | | DermNet-S (8 classes) | |
|---|---|---|---|---|---|
| | | 1-shot | 5-shot | 1-shot | 5-shot |
| | SL | $32.37 \pm 0.60\%$ | $45.57 \pm 0.69\%$ | $35.69 \pm 0.58\%$ | $53.38 \pm 0.60\%$ |
| | MTL | $33.01 \pm 0.64\%$ | $46.79 \pm 0.65\%$ | $36.20 \pm 0.64\%$ | $54.53 \pm 0.63\%$ |
| MAML | Vanilla | $36.09 \pm 0.75\%$ | $50.01 \pm 0.67\%$ | $37.98 \pm 0.66\%$ | $54.35 \pm 0.67\%$ |
| | MetaMix | $37.74 \pm 0.77\%$ | $51.79 \pm 0.68\%$ | $40.36 \pm 0.73\%$ | $55.75 \pm 0.69\%$ |
| | **MLTI (ours)** | $38.13 \pm 0.70\%$ | $\mathbf{53.53 \pm 0.72\%}$ | $\mathbf{41.32 \pm 0.71\%}$ | $\mathbf{56.95 \pm 0.64\%}$ |
| ProtoNet | Vanilla | $35.07 \pm 0.73\%$ | $45.10 \pm 0.63\%$ | $37.72 \pm 0.67\%$ | $53.18 \pm 0.66\%$ |
| | MetaMix | $38.12 \pm 0.71\%$ | $50.25 \pm 0.69\%$ | $40.07 \pm 0.69\%$ | $55.07 \pm 0.68\%$ |
| | **MLTI (ours)** | $\mathbf{39.64 \pm 0.77\%}$ | $51.64 \pm 0.65\%$ | $41.31 \pm 0.71\%$ | $56.09 \pm 0.67\%$ |

Table 14: Results on Huffpost and tieredImageNet-S. Here, averaged accuracies $\pm$ 95% confidence intervals are reported.

| Backbone | Strategies | tieredImageNet-S | | NLP: Huffpost | |
|---|---|---|---|---|---|
| | | 1-shot | 5-shot | 1-shot | 5-shot |
| MAML | Vanilla | $42.20 \pm 0.84\%$ | $58.23 \pm 0.77\%$ | $39.51 \pm 1.07\%$ | $50.68 \pm 0.90\%$ |
| | Meta-Reg | $42.87 \pm 0.86\%$ | $59.16 \pm 0.79\%$ | $40.32 \pm 1.05\%$ | $50.96 \pm 0.98\%$ |
| | TAML | $42.86 \pm 0.84\%$ | $59.33 \pm 0.76\%$ | $40.03 \pm 1.00\%$ | $50.89 \pm 0.88\%$ |
| | Meta-Dropout | $41.94 \pm 0.82\%$ | $58.37 \pm 0.77\%$ | $39.89 \pm 0.98\%$ | $51.03 \pm 0.91\%$ |
| | MetaMix | $43.40 \pm 0.85\%$ | $61.92 \pm 0.80\%$ | $40.64 \pm 1.02\%$ | $51.65 \pm 0.92\%$ |
| | Meta-Maxup | $43.69 \pm 0.88\%$ | $60.00 \pm 0.82\%$ | $40.39 \pm 1.01\%$ | $51.80 \pm 0.91\%$ |
| | **MLTI (ours)** | $\mathbf{44.32 \pm 0.82\%}$ | $\mathbf{62.22 \pm 0.79\%}$ | $\mathbf{41.06 \pm 1.04\%}$ | $\mathbf{52.53 \pm 0.90\%}$ |
| ProtoNet | Vanilla | $43.35 \pm 0.82\%$ | $59.98 \pm 0.77\%$ | $41.85 \pm 1.01\%$ | $58.98 \pm 0.92\%$ |
| | MetaMix | $44.14 \pm 0.83\%$ | $60.97 \pm 0.81\%$ | $42.27 \pm 0.98\%$ | $60.43 \pm 0.90\%$ |
| | Meta-Maxup | $44.40 \pm 0.83\%$ | $61.79 \pm 0.78\%$ | $42.39 \pm 1.01\%$ | $60.27 \pm 0.88\%$ |
| | **MLTI (ours)** | $\mathbf{45.47 \pm 0.86\%}$ | $\mathbf{62.35 \pm 0.80\%}$ | $\mathbf{42.74 \pm 0.96\%}$ | $\mathbf{61.09 \pm 0.91\%}$ |

Table 15: Full table of the overall performance (averaged accuracy $\pm$ 95% confidence interval) under the non-label-sharing scenario.

| Backbone | Strategies | miniImagenet-S | ISIC | DermNet-S | Tabular Murris |
|---|---|---|---|---|---|
| MAML (1-shot) | Vanilla | $38.27 \pm 0.74\%$ | $57.59 \pm 0.79\%$ | $43.47 \pm 0.83\%$ | $79.08 \pm 0.91\%$ |
| | Meta-Reg | $38.35 \pm 0.76\%$ | $58.57 \pm 0.94\%$ | $45.01 \pm 0.83\%$ | $79.18 \pm 0.87\%$ |
| | TAML | $38.70 \pm 0.77\%$ | $58.39 \pm 1.00\%$ | $45.73 \pm 0.84\%$ | $79.82 \pm 0.87\%$ |
| | Meta-Dropout | $38.32 \pm 0.75\%$ | $58.40 \pm 1.02\%$ | $44.30 \pm 0.84\%$ | $78.18 \pm 0.93\%$ |
| | MetaMix | $39.43 \pm 0.77\%$ | $60.34 \pm 1.03\%$ | $46.81 \pm 0.81\%$ | $81.06 \pm 0.86\%$ |
| | Meta-Maxup | $39.28 \pm 0.77\%$ | $58.68 \pm 0.86\%$ | $46.10 \pm 0.82\%$ | $79.56 \pm 0.89\%$ |
| | **MLTI (ours)** | $\mathbf{41.58 \pm 0.72\%}$ | $\mathbf{61.79 \pm 1.00\%}$ | $\mathbf{48.03 \pm 0.79\%}$ | $\mathbf{81.73 \pm 0.89\%}$ |
| MAML (5-shot) | Vanilla | $52.14 \pm 0.65\%$ | $65.24 \pm 0.77\%$ | $60.56 \pm 0.74\%$ | $88.55 \pm 0.60\%$ |
| | Meta-Reg | $51.74 \pm 0.68\%$ | $68.45 \pm 0.81\%$ | $60.92 \pm 0.69\%$ | $89.08 \pm 0.61\%$ |
| | TAML | $52.75 \pm 0.70\%$ | $66.09 \pm 0.71\%$ | $61.14 \pm 0.72\%$ | $89.11 \pm 0.59\%$ |
| | Meta-Dropout | $52.53 \pm 0.69\%$ | $67.32 \pm 0.92\%$ | $60.86 \pm 0.73\%$ | $89.25 \pm 0.59\%$ |
| | MetaMix | $54.14 \pm 0.73\%$ | $69.47 \pm 0.60\%$ | $63.52 \pm 0.73\%$ | $89.75 \pm 0.58\%$ |
| | Meta-Maxup | $53.02 \pm 0.72\%$ | $69.16 \pm 0.61\%$ | $62.64 \pm 0.72\%$ | $88.88 \pm 0.57\%$ |
| | **MLTI (ours)** | $\mathbf{55.22 \pm 0.76\%}$ | $\mathbf{70.69 \pm 0.68\%}$ | $\mathbf{64.55 \pm 0.74\%}$ | $\mathbf{91.08 \pm 0.54\%}$ |
| ProtoNet | Vanilla | $36.26 \pm 0.70\%$ | $58.56 \pm 1.01\%$ | $44.21 \pm 0.75\%$ | $80.03 \pm 0.90\%$ |
| | MetaMix | $39.67 \pm 0.71\%$ | $60.58 \pm 1.17\%$ | $47.71 \pm 0.83\%$ | $80.72 \pm 0.90\%$ |
| | Meta-Maxup | $39.80 \pm 0.73\%$ | $59.66 \pm 1.13\%$ | $46.06 \pm 0.78\%$ | $80.87 \pm 0.95\%$ |
| | **MLTI (ours)** | $\mathbf{41.36 \pm 0.75\%}$ | $\mathbf{62.82 \pm 1.13\%}$ | $\mathbf{49.38 \pm 0.85\%}$ | $\mathbf{81.89 \pm 0.88\%}$ |
| ProtoNet | Vanilla | $50.72 \pm 0.70\%$ | $66.25 \pm 0.96\%$ | $60.33 \pm 0.70\%$ | $89.20 \pm 0.56\%$ |
| | MetaMix | $53.10 \pm 0.74\%$ | $70.12 \pm 0.94\%$ | $62.68 \pm 0.71\%$ | $89.30 \pm 0.61\%$ |
| | Meta-Maxup | $53.35 \pm 0.68\%$ | $68.97 \pm 0.83\%$ | $62.97 \pm 0.74\%$ | $89.42 \pm 0.64\%$ |
| | **MLTI (ours)** | $\mathbf{55.34 \pm 0.74\%}$ | $\mathbf{71.52 \pm 0.89\%}$ | $\mathbf{65.19 \pm 0.73\%}$ | $\mathbf{90.12 \pm 0.59\%}$ |

Table 16: Full table (accuracy $\pm$ 95% confidence interval) of the cross-domain adaptation under the non-label-sharing scenario. A $\rightarrow$ B represents that the model is meta-trained on A and then is meta-tested on B.

| Model | | miniImagenet-S $\rightarrow$ Dermnet-S | | Dermnet-S $\rightarrow$ miniImagenet-S | |
|---|---|---|---|---|---|
| | | 1-shot | 5-shot | 1-shot | 5-shot |
| MAML | | $33.67 \pm 0.61\%$ | $50.40 \pm 0.63\%$ | $28.40 \pm 0.55\%$ | $40.93 \pm 0.63\%$ |
| | +MLTI | $\mathbf{36.74 \pm 0.64\%}$ | $\mathbf{52.56 \pm 0.62\%}$ | $\mathbf{30.03 \pm 0.58\%}$ | $\mathbf{42.25 \pm 0.64\%}$ |
| ProtoNet | | $33.12 \pm 0.60\%$ | $50.13 \pm 0.65\%$ | $28.11 \pm 0.53\%$ | $40.35 \pm 0.61\%$ |
| | +MLTI | $\mathbf{35.46 \pm 0.63\%}$ | $\mathbf{51.79 \pm 0.62\%}$ | $\mathbf{30.06 \pm 0.56\%}$ | $\mathbf{42.23 \pm 0.61\%}$ |

