# OpenReview forum: "Meta-Learning with Fewer Tasks through Task Interpolation"
_ICLR.cc/2022/Conference — ICLR 2022 Oral_

### Official Review · Reviewer_id95 · 2021-10-29

**Correctness:** 3
**Technical Novelty And Significance:** 3
**Empirical Novelty And Significance:** 3
**Recommendation:** 8
**Confidence:** 3

**Main Review:**

I found the approach to be simple and relatively well explained, including ablations studies on large-point questions I had while reading, including its behavior and effectiveness for different sizes and number of classes in the original base dataset, as well as effects of inter- and intra- task interpolations.

The key difference between this work and MetaMix (Yao et al 2020) is incremental but important:  MetaMix will run the inner loop on the unmodified support set only, and use a mix of support+query in outer loop comparison optimization, whereas this work interpolates support set in inner loop as well.  This difference enables between-task interpolation which adds additional augmentation particularly in settings where few tasks can be drawn from the base data.

I didn't follow much of the theoretical sections in detail, and had to look at the appendix proofs to even understand some of the notation in the main text.  In my somewhat limited understanding they seem reasonable.  These claim to show a theoretical generalization improvement in simplified settings (binary classification of single layer model, linear protonet feautres).


Additional questions:

NLS:  In addition to a single set of correspondence pairs, the input examples for each class can be mixed with all-pairwise-combinations.  How many combinations are used?  That is, for two sets of k examples {xs_i} and {xq_j} (i,j in 1..k), one can form k^2 interpolated examples {a xs_i + (1-a) xq_j} using each i,j combination.  Are all of these combinations formed or just a single set of k pairings?  If using more than k pairings, this would change the task from k-shot to k^2-shot; but the l-layer features for each of the k^2 combinations could be computed, and then up to (k^2 choose k) tasks could be selected from these and used in the upper layer loss comparisons.  For k=5 that would increase interpolated pairs from 5 to 25, but potentially get up to 53130 upper layer loss comparisons from each task pair sample -- would this get even more benefit from this task augmentation technique?

eq 5:  what does the name of the subscript "cr" mean (does it stand for something)?

It could be useful to have a more explicit explanation of differences with MetaMix.  MetaMix will run the inner loop on the unmodified support set only, and use a mix of support+query in outer loop comparison optimization, whereas this work interpolates support set in inner loop as well.  This is already mentioned at a high level (fig 1 caption and sec 5 last paragraph), but I think it could be even clearer by pointing out the difference in the discussion around eq 5, that support set H^s,Y^s in the inner loop is mixed between tasks, whereas in MetaMix, only the H^q,Y^q are replaced by mixing.



**Summary Of The Paper:**

This paper describes a method for augmenting task selection in meta-learning, by interpolating support and query sets between two random tasks from the base dataset.  This is examined in two scenarios, label-shared LS and non-label-shared NLS, differing in whether the label space is the same between tasks (e.g. pose estimation) or different (classification to different discrete class sets).  In the former, label targets are interpolated as well as support set inputs, while in the latter, new classes are constructed by random cross-task pairings.  Comparisons are made to other interpolation augmentation approaches, including MetaMix, which interpolates in query set but not the support set.  The approach results in significant performance gains on multiple benchmarks in both settings.


**Summary Of The Review:**

Overall, the approach is described well enough to understand the approach, and is emperically shown to result in decent performance gains in the low task data settings for which it is intended.  The theoretical sections corroborate this, but I found them hard to follow.

---

> ### Author Response · Authors · 2021-11-16
> **Response to Reviewer id95**
>
> We really appreciate your constructive suggestions to improve the paper. Below, we address your concerns point by point. Please kindly let us know if your concerns are addressed and whether you have any further concerns.
>
>
> **Q1**: Theoretical Analysis
>
> **A1**: Thanks for pointing this out. We have revised the theoretical part by adding more explanations. Specifically, we’ve now stated more explicitly that the loss function induced by MLTI is approximately the standard loss function plus a positive term. We explained that such a positive term is a quadratic function on the parameter phi_i’s, and therefore can be interpreted as a regularization term. Furthermore, we added more explanations around Theorem 1, and indicated that the purpose of Theorem is to show how this regularization term improves the generalization error bound. Please kindly let us know if you have any additional questions.
>
>
> ---
>
> **Q2**: Sample selection in NLS setting
>
> **A2**: Thank you for your comment. In our paper, for each interpolated task, only k pairings are used as the new support set, and thus it is still k-shot learning. To be more specific, assume we have two sets (S1 and S2) to be interpolated. We first shuffle samples in S2 and then interpolate the shuffled S2 with S1.
>
> We conduct the experiments by allowing all-pairwise combinations and report the results of miniImagenet-S and Dermnet-S 1-shot setting in Table R-8.
>
> **Table R-8**: Comparison between MLTI and MLTI with all-pairwise combinations
>
> |                |    miniImagenet-S   |      Dermnet-S      |
> |----------------|:-------------------:|:-------------------:|
> | MLTI           | 41.58 $\pm$ 0.72% | 48.03 $\pm$ 0.79% |
> | MLTI-all pairs | 41.72 $\pm$ 0.75% | 48.07 $\pm$ 0.82% |
>
>
> From the results, we find that using more pairs does not provide much benefit to the predictive accuracy. One possible reason for this is that the current combinations have already made use of most of the essential information in the training data for this method. A similar phenomenon has also been observed in other data augmentation techniques, e.g., using bootstrap can hypothetically generate infinitely many new samples. But in practice, people find that generating the same size as the original training data is sufficient. Similarly, the vanilla mixup, although one can again generate infinitely many new samples, people commonly only generate the same amount of new samples as the training data and find it performs well in practice.
>
> ---
>
> **Q3**: subscript "cr" in eq5
>
> **A3**: cr represents “cross”. We have revised the paper to clarify this.
>
> ---
>
> **Q4**: Point out the difference between MetaMix in eq5
>
> **A4**: Thanks for the suggestion, we have revised our descriptions around eq5 and explicitly point out the differences.

---

### Official Review · Reviewer_A6CS · 2021-11-01

**Correctness:** 4
**Technical Novelty And Significance:** 3
**Empirical Novelty And Significance:** 3
**Recommendation:** 8
**Confidence:** 4

**Main Review:**

Strength
1.	This paper proposes a novel task-augmentation method, which is affected by Manifold Mixup, which can be applied to many existing few-shot learning tasks.
2.	The theoretical analysis shows that the proposed MLTI augmentation has a regularization effect and leads the meta-learner to have a better generalization capability.
3.	Extensive simulation results on variety of few-shot learning datasets and two representative few-shot learning methods show that the proposed MLTI is highly effective for meta-learning with fewer data.

Weakness
1.	Comparison with the prior methods in large dataset is missing. For example, in Table 3, the comparison results are provided only for small datasets or reduced version of large datasets. However, the proposed method is not restricted to small dataset. The ablation experimental result in Figure 2 shows that proposed MLTI is still effective when the full miniImageNet/DermNet dataset is used, although the performance gain becomes small when the full dataset is used. I suggest the authors to include the comparison of MLTI and prior methods with full size of miniImageNet and DermNet.

Question
1.	In Section 3, the authors mentioned that it is intractable to calculate prototypes with mixed labels. However, in prior work on semi-supervised few-shot learning [1], the prototypes are computed using soft-labels. What happens if we compute prototypes using soft-labels as done in [1]?
2.	Some additional studies on the interpolation layer would be helpful for understanding the proposed method. In Algorithm 1 and 3, the interpolation layer $l$ is randomly chosen in step 7. What happens if we fix $l$ instead of randomly sampling $l$ for every iteration? In that case, how are interpolating at lower layer and interpolating higher layer different?

Typo: In last line of page 4, there is a typo (regularizaiton -> regularization)

[1] Ren, Mengye, et al. "Meta-Learning for Semi-Supervised Few-Shot Classification." International Conference on Learning Representations. 2018.


**Summary Of The Paper:**

This paper proposes a task augmentation method via task-interpolation for data efficient meta-learning. While the traditional meta-learning methods highly rely on a large amount of data to retain diverse training tasks, the proposed method, MLTI generates tasks by interpolating the tasks which are obtained from training data. The experimental results on variety of few-shot learning dataset show that MLTI is effective when the meta-training data for constructing training tasks is not enough, for both gradient-based and metric-based few-shot learner.

**Summary Of The Review:**

This paper proposes a novel task-augmentation method of MLTI and shows the effectiveness of proposed MLTI through the extensive simulation results and theoretical analysis. The proposed MLTI can be applied to both optimization-based and metric-based few-shot learning methods. Adding some experimental results would make the readers to better understand the proposed method.
However, I believe the idea of this paper is valuable for few-shot learning field, and I recommend to accept this paper.

---

> ### Author Response · Authors · 2021-11-16
> **Response to Reviewer A6CS**
>
> Thank you for reviewing our paper and your constructive comments. We have fixed the typos in the paper, and we provide detailed responses below. Would you mind letting us know if our responses address your concerns?
>
> **Q1**: Full-size miniImagenet and DermNet.
>
> **A1**: Thanks for the great suggestion; we have added the results of all baselines on full-size miniImagenet and DermNet, which show the effectiveness of MLTI. Please kindly refer to Table R-5 and Appendix E.2 in the revised paper. The results further demonstrate the effectiveness of MLTI. All results are calculated on the same meta-testing tasks.
>
> **Table R-5**: Results on full-size miniImagenet and DermNet
>
>  | Backbone | Strategies   | miniImagenet-full |                   | DermNet-full      |                   |
> |----------|--------------|:-----------------:|:-----------------:|:-----------------:|:-----------------:|
> |          |              |       1-shot      |       5-shot      | 1-shot            | 5-shot            |
> | MAML     | Vanilla      | 46.90 $\pm$ 0.79% | 63.02 $\pm$ 0.68% | 49.58 $\pm$ 0.83% | 69.15 $\pm$ 0.69% |
> |          | Meta-Reg     | 47.02 $\pm$ 0.77% | 63.19 $\pm$ 0.69% | 50.10 $\pm$ 0.86% | 69.73 $\pm$ 0.70% |
> |          | TAML         | 46.40 $\pm$ 0.82% | 63.26 $\pm$ 0.68% | 50.26 $\pm$ 0.85% | 69.40 $\pm$ 0.75% |
> |          | Meta-Dropout | 47.47 $\pm$ 0.81% | 64.11 $\pm$ 0.71% | 51.10 $\pm$ 0.84% | 69.08 $\pm$ 0.69% |
> |          | MetaMix      | 47.81 $\pm$ 0.78% | 64.22 $\pm$ 0.68% | 51.83 $\pm$ 0.83% | 71.57 $\pm$ 0.67% |
> |          | Meta-Maxup   | 47.68 $\pm$ 0.79% | 63.51 $\pm$ 0.75% | 51.95 $\pm$ 0.88% | 70.84 $\pm$ 0.68% |
> |          | **MLTI (ours)**         | **48.62 $\pm$ 0.76%** | **64.65 $\pm$ 0.70%** | **52.32 $\pm$ 0.88%** | **71.77 $\pm$ 0.67%** |
> | ProtoNet | Vanilla      | 47.05 $\pm$ 0.79% | 64.03 $\pm$ 0.68% | 49.91 $\pm$ 0.79% | 67.45 $\pm$ 0.70% |
> |          | MetaMix      | 47.21 $\pm$ 0.76% | 64.38 $\pm$ 0.67% | 51.50 $\pm$ 0.76% | 69.55 $\pm$ 0.68% |
> |          | Meta-Maxup   | 47.33 $\pm$ 0.79% | 64.43 $\pm$ 0.69% | 51.18 $\pm$ 0.83% | 69.07 $\pm$ 0.72% |
> |          | **MLTI (ours)**         | **48.11 $\pm$ 0.81%** | **65.22 $\pm$ 0.70%** | **52.91 $\pm$ 0.81%** | **71.30 $\pm$ 0.69%** |
>
> ---
>
> **Q2**: soft-label to calculate the prototype.
>
> **A2**: Thank you for pointing out this idea to us. We tried the idea to compute prototypes using soft-labels on RainbowMNIST. Specifically, we use the interpolation ratio $\lambda$ as soft labels to calculate the prototypes. The results are reported in Table R-6:
>
> **Table R-6**: Comparison of MLTI w/ or w/o using soft labels to construct prototypes
>
> | Strategies      |    RainbowMNIST   |
> |-----------------|:-----------------:|
> | Vanilla         | 65.41 $\pm$ 1.10% |
> | MLTI+Soft label | 59.34 $\pm$ 0.99% |
> | **MLTI (ours)**           | **70.14 $\pm$ 0.92%** |
>
> From these results, we found that using soft labels to construct prototypes performs significantly worse than MLTI and Vanilla ProtoNet. We speculate this result is due to the prototype computed using soft-labels, combined with the mixing idea, which cannot distinguish between the different classes well enough. In the vanilla setting without mixing, each prototype has the interpretation to represent its own class. Now, with mixing (for simplicity, let’s consider a fixed non-zero mixing proportion lambda), each prototype contains a non-trivial proportion of information from more than one class, making it harder to distinguish between the different classes.
>
> ---
>
> **Q3**: Interpolation layer analysis in Manifold Mixup
>
> **A3**: We conduct new experiments on Metabolism and Tabular Murris, where Manifold Mixup (i.e., interpolating features) is used for sample interpolation, where ProtoNet is used as backbone algorithm. We report the results in Table R-7 and Appendix E.5.2. The results indicate (1) fixing the interpolation layer can also boost the performance; (2) randomly selecting the interpolation layer achieves the best performance; (3) interpolating at the lower layer performs similarly as interpolating at the higher layer, indicating the robustness of MLTI with different selected layers.
>
> **Table R-7**: Analysis of interpolation layers. Layer 0, 1 represents randomly select layer 0 or layer 1 for interpolation. None means vanilla ProtoNet.
>
> |      Interpolation Layer      | Metabolism: 5-shot | Tabular Murris: 1-shot |
> |------------|:------------------:|:----------------------:|
> | None | 61.06 $\pm$ 0.94% | 80.03 $\pm$ 0.90% |
> | Layer 0    |  62.53 $\pm$ 0.98% |    81.18 $\pm$ 0.93%   |
> | Layer 1    |  62.38 $\pm$ 0.94% |    81.25 $\pm$ 0.90%   |
> | **Layer 0, 1** |  **63.47 $\pm$ 0.96%** |    **81.89 $\pm$ 0.88%**   |

---

### Official Review · Reviewer_hkv8 · 2021-11-01

**Correctness:** 3
**Technical Novelty And Significance:** 3
**Empirical Novelty And Significance:** 3
**Recommendation:** 8
**Confidence:** 3

**Main Review:**

strengths

Although nifty, the idea of pair-wise task interpolation is an incremental change over the existing data augmentation approaches. The theoretical results, highlighting the relationship between task interpolation and the Rademacher complexity, are non-trivial extensions of the Zhang et al. ICLR 2021 and Yao et al. ICML2021 to account for pair-wise task interpolation. I view this as the primary contribution of the paper.

The comparison against existing data-augmentation baselines for both metric- and gradient-based meta-learning approaches is quite exhaustive. Furthermore, MLTI is tested on a wide variety of datasets. While the improvement on each dataset is only marginal, the consistent improvement in all datasets and across all approaches strengthens the paper’s contribution.

The paper is well-written and easy to follow.

Concerns

Current approaches in meta-learning rely on heavier backbones such as ResNet. As the goal of all the meta-learning methods is to improve the model’s generalizability, I think it is fair to evaluate the effectiveness of MLTI with heavier feature extraction backbones. Such a comparison is relevant as the proposed task interpolation is conducted on the features extracted from some intermediate layer of the network.


**Summary Of The Paper:**

The paper proposes an interpolation strategy for meta-learning to improve the learned model’s generalizability. The interpolation strategy is quite simple - interpolate between a pair of tasks, in contrast to existing methods such as adding label noise or data augmentation on each task individually. The authors show that the resulting gradient- or metric-based meta-learning framework (MLTI) induces a data-dependent regularizer that controls the Rademacher complexity leading to better generalization. MLTI is tested on specially curated datasets derived from standard benchmarking datasets. Furthermore, MLTI is also compared against existing data augmentation and interpolation strategies for meta-learning to illustrate its effectiveness.


**Summary Of The Review:**

Overall, the paper proposes a simple extension to standard data/task - augmentation methods for meta-learning but justifies it with rigorous theory. The theoretical results are non-trivial extensions/combinations of existing work. The effectiveness of the approach is evident from the extensive empirical evaluation. The contributions are strong, albeit limited to the meta-learning research community.

---

> ### Author Response · Authors · 2021-11-16
> **Response to Reviewer hkv8**
>
> Thanks a lot for reviewing our paper and acknowledging our contributions. We have conducted additional experiments as suggested and please kindly let us know if your concerns are addressed.
>
> **Q1**: Heavier feature extraction backbone.
>
> **A1**: We ran a new experiment with ResNet-12 as the backbone and evaluated the performance of 1-shot miniImagenet-S and DermNet-S with this heavier backbone. The results are reported in the following Table R-4 and in Appendix E.5.1. The results demonstrate the consistent effectiveness of MLTI even with deeper backbone models.
>
> **Table R-4**: Results of MLTI and other strategies with the heavier base model (ResNet-12)
>
> | Backbone | Strategies  |    miniImagenet-S  |      DermNet-S      |
> |----------|-------------|:-----------------:|:-----------------:|
> | MAML     | Vanilla     | 40.02 $\pm$ 0.78% | 47.58 $\pm$ 0.93% |
> |          | MetaMix     | 42.26 $\pm$ 0.75% | 51.40 $\pm$ 0.89% |
> |          | Meta-Maxup  | 41.97 $\pm$ 0.78% | 50.82 $\pm$ 0.85% |
> |          | **MLTI (ours)** | **43.35 $\pm$ 0.80%** | **52.03 $\pm$ 0.90%** |
> | ProtoNet | Vanilla     | 40.96 $\pm$ 0.75% | 48.65 $\pm$ 0.85% |
> |          | MetaMix     | 42.95 $\pm$ 0.87% | 51.18 $\pm$ 0.90% |
> |          | Meta-Maxup  | 42.68 $\pm$ 0.78% | 50.96 $\pm$ 0.88% |
> |          | **MLTI (ours)** | **44.08 $\pm$ 0.83%** | **52.01 $\pm$ 0.93%** |

---

### Official Review · Reviewer_maUk · 2021-11-02

**Correctness:** 3
**Technical Novelty And Significance:** 3
**Empirical Novelty And Significance:** 3
**Recommendation:** 8
**Confidence:** 4

**Main Review:**

[Strengths]
The work provides extensive theoretical analysis to provide theoretical guarantees as to how the proposed MLTI task interpolation method achieves better generalization. In contrast, previous methods that have employed common augmentation methods (e.g., label noise, CutMix, MixUp) without theoretical guarantees.
The work introduces scenarios that are more challenging than standard benchmarks by limiting the number of meta-training tasks.
The work provides extensive experiments across various datasets under such challenging scenarios and demonstrates better performance than previous methods, providing empirical support for the effectiveness of the proposed task interpolation method.

[Weaknesses]
I believe the work has minor technical novelty compared to the related work by Ni et al [1]. In particular, Ni et al. [1] performs several augmentations for meta-learning, one of which is MixUp for tasks. Using MixUp between any given pair of classes, Ni et al. [1] also creates new tasks.

[Comments]
In Related Work section, the work states that compared to work by Ni et al. [1], the proposed method directly desnifies the task distribution. But, doesn’t [1] effectively densify the task distribution, where a new task can consist of new classes that are constructed by using MixUp on pairs of classes? As such, I believe more discussions on this issue should help better differentiate the work from the related work.
Why does the proposed method randomly sample a location where features are to be interpolated? Is there an ablation study on the sampled location? I wonder if this technique is what makes the proposed method perform better than other works. I’m curious as to whether the proposed method, without this technique, still performs better than other works. The ablation study on this would be helpful for better understanding of differences from other works.
Also, how does it compare with related works on standard benchmarks, such as miniImageNet. I think that the proposed method should still work with a larger number of tasks and believe that these experimental comparisons can strengthen the contributions of the proposed method.

[1] Ni et al. Data Augmentation for Meta-Learning.

**Summary Of The Paper:**

[Summary]
This work tackles a scenario where there may not be a large number of training tasks available, which increases the susceptibility of meta-learning algorithms to meta-level overfitting/memorization problem. In particular, to cope with the scarcity of tasks, the paper proposes to augment the given task set through interpolation of tasks. The paper reports better performance than other methods on benchmarks that have fewer training tasks.


**Summary Of The Review:**

[Recommendation]
Despite strong experimental results and analysis, at this point, I believe the technical novelties are not significantly different from the work by Ni et al. Thus, I believe the work is marginally below the acceptance threshold. If the above comments are addressed, I’m willing to increase the score.

-----------------------------------------
[Post-rebuttal]
I thank the authors for the response, along with clarifications and updates in the manuscript.
As the authors have addressed most of my concerns, I'm happy to increase the score accordingly.

---

> ### Author Response · Authors · 2021-11-16
> **Response to Reviewer maUk (1/2)**
>
> Thank you for your review and for acknowledging our empirical studies. We respond to your concerns below. We would really appreciate it if you could let us know whether your concerns are addressed by the response.
>
> **Q1**: Difference between MLTI and Meta-Maxup [Ni et al. 2021]
>
> **A1**: Below, we elaborate on the differences between individual task interpolation like Meta-Maxup [Ni et al. 2021] and MLTI. We have revised the paper to include these clarifications in Appendix C. The reasons why MLTI leads to more dense task distribution could be summarized under both label-sharing and non-label-sharing settings:
>
> - Under the label-sharing setting, MLTI densifies the task distribution by enabling cross-task interpolation. For example, in Pose prediction, we not only interpolate samples within each object, but cross-task interpolation significantly increases the number of tasks. Assume we have two objects (O1 and O2), individual task interpolation approaches (e.g., Meta-Maxup) only generate more samples in O1 or O2, where only one object information is covered. However, MLTI further allows generating tasks with both O1 and O2 information by interpolating data samples from O1 and O2.
>
> - Under the non-label-sharing setting, MLTI also leads to more dense task distribution. For example, in 2-way classification with 3 training classes (C0, C1, C2), there are three original tasks, i.e., three classification pairs (C0, C1), (C0, C2), (C1, C2). Individual task interpolation increases the number of samples for each classification pair by enabling data from mix(C0, C1), mix(C0, C2), mix(C1, C2). However, it does not distinguish pairs like (mix(C0, C1), mix(C0, C2)), whereas MLTI does by allowing cross-tasks interpolation.
>
> Finally, it is also worthwhile to mention that MLTI empirically outperforms individual task interpolation methods -- MetaMix and Meta-Maxup.
>
> ---
>
> **Q2**: Randomly sampled location
>
> **A2**: As mentioned under equation (5), we use Manifold Mixup for task interpolation, where the interpolation layer (location) is randomly selected as suggested in [Verma et al. 2019]. We also empirically find that the randomly selected interpolation layer achieves the best performance when Manifold Mixup is used for task interpolation. The paper also mentions that Manifold Mixup can be replaced with other interpolation strategies (e.g., CutMix, mixup) under equation (5) and the selection of interpolation strategy has been discussed in Appendix D.1 and E.1.
>
> Most importantly, all interpolation-based models -- MetaMix, Meta-Maxup, MLTI use the same interpolation strategies for all experiments, i.e., randomly sampling location is used for all interpolation-based strategies when Manifold Mixup is adopted in the experiments. We have revised the paper to include these details in Appendix D.1 and E.1. Thus, the performance gains of MLTI over other strategies are not caused by the random sample strategy.

---

> > ### Author Response · Authors · 2021-11-16
> > **Response to Reviewer maUk (2/2)**
> >
> > **Q3**: Results on full-size image data
> >
> > **A3**: The original submission already shows the performance of MLTI under the full-size miniImagenet and DermNet in Figure 2, where miniImagenet with 64 training classes and DermNet with 150 training classes are full-size datasets. All results are reported with the same meta-testing tasks. We further provide the comparison between MLTI and other baseline strategies in the following Table R-3 and Appendix E.2 (Table 8), where MLTI also shows its promise even under the full-size setting.
> >
> > **Table R-3**: Results on full-size miniImagenet and DermNet
> >
> >  | Backbone | Strategies   | miniImagenet-full |                   | DermNet-full      |                   |
> > |----------|--------------|:-----------------:|:-----------------:|:-----------------:|:-----------------:|
> > |          |              |       1-shot      |       5-shot      | 1-shot            | 5-shot            |
> > | MAML     | Vanilla      | 46.90 $\pm$ 0.79% | 63.02 $\pm$ 0.68% | 49.58 $\pm$ 0.83% | 69.15 $\pm$ 0.69% |
> > |          | Meta-Reg     | 47.02 $\pm$ 0.77% | 63.19 $\pm$ 0.69% | 50.10 $\pm$ 0.86% | 69.73 $\pm$ 0.70% |
> > |          | TAML         | 46.40 $\pm$ 0.82% | 63.26 $\pm$ 0.68% | 50.26 $\pm$ 0.85% | 69.40 $\pm$ 0.75% |
> > |          | Meta-Dropout | 47.47 $\pm$ 0.81% | 64.11 $\pm$ 0.71% | 51.10 $\pm$ 0.84% | 69.08 $\pm$ 0.69% |
> > |          | MetaMix      | 47.81 $\pm$ 0.78% | 64.22 $\pm$ 0.68% | 51.83 $\pm$ 0.83% | 71.57 $\pm$ 0.67% |
> > |          | Meta-Maxup   | 47.68 $\pm$ 0.79% | 63.51 $\pm$ 0.75% | 51.95 $\pm$ 0.88% | 70.84 $\pm$ 0.68% |
> > |          | **MLTI (ours)**         | **48.62 $\pm$ 0.76%** | **64.65 $\pm$ 0.70%** | **52.32 $\pm$ 0.88%** | **71.77 $\pm$ 0.67%** |
> > | ProtoNet | Vanilla      | 47.05 $\pm$ 0.79% | 64.03 $\pm$ 0.68% | 49.91 $\pm$ 0.79% | 67.45 $\pm$ 0.70% |
> > |          | MetaMix      | 47.21 $\pm$ 0.76% | 64.38 $\pm$ 0.67% | 51.50 $\pm$ 0.76% | 69.55 $\pm$ 0.68% |
> > |          | Meta-Maxup   | 47.33 $\pm$ 0.79% | 64.43 $\pm$ 0.69% | 51.18 $\pm$ 0.83% | 69.07 $\pm$ 0.72% |
> > |          | **MLTI (ours)**         | **48.11 $\pm$ 0.81%** | **65.22 $\pm$ 0.70%** | **52.91 $\pm$ 0.81%** | **71.30 $\pm$ 0.69%** |
> >
> > ---
> >
> > **Reference**
> >
> > [Ni et al. 2021] Ni, Renkun, Micah Goldblum, Amr Sharaf, Kezhi Kong, and Tom Goldstein. "Data augmentation for meta-learning." ICML 2021.

---

### Official Review · Reviewer_a9eU · 2021-11-02

**Correctness:** 4
**Technical Novelty And Significance:** 3
**Empirical Novelty And Significance:** 2
**Recommendation:** 8
**Confidence:** 3

**Main Review:**

### Pros
+ The paper is well written and easy to follow.
+ The idea of interpolating tasks in a meta-learning setting is novel, intuitive and simple.  Although previous work exists that augment the number of tasks, this is the first approach that augments across tasks (rather than within task).
+ The authors shows good result on different datasets, settings and backbones.

### Cons
- I feel like results in more “traditional” (and larger) FSL datasets are missing. For example, it would be nice to see results in tieredImagerNet or metaDataset.
- I also feel that the authors introduce the method as being a general meta-learning approach, but only show results on image. classification/regression. It would be nice to see results in other domains such as RL/NLP/etc tasks.
- I find the theoretical analysis difficult to follow and potentially not very informative to the rest of the paper (that been said, I am not an expert on generalization theory/Rademacher complexity and cannot properly validate it).

**Summary Of The Paper:**

In this paper the authors propose a meta-learning method for few-shot learning. The propose approach, MLTI, creates new (artificial) tasks by interpolating two (existing) tasks form the training set during (meta-)training. The new tasks are generated by interpolating features/labels of two sampled tasks from the training set. The authors show that the proposed approach achieve good results in multiple datasets (both in regression and classification), multiple settings (”label sharing” and “non-label sharing”) and different algorithms (MAML and ProtoNets).


**Summary Of The Review:**

I recommend this paper for acceptance. The proposed idea is simple and novel, the paper is well written and the empirical evaluation is well executed.

---
**Post-rebuttal update**

I thank the reviewers for the rebuttal and I keep my rating of 8.
Congratulations for the nice work!

---

> ### Author Response · Authors · 2021-11-16
> **Response to Reviewer a9eU**
>
> Thank you for reviewing the paper and your valuable comments and suggestions to improve the paper. We detail our response below and have revised the paper accordingly. Please kindly let us know whether our response addresses your concerns.
>
> **Q1**: Results on traditional FSL datasets
>
> **A1**: We have conducted experiments on tieredImageNet, and list the results in Table R-1 and in Appendix E.8 (Table 14) of the revised paper. Similar to miniImagenet and DermNet, we apply MLTI on tieredImageNet with fewer classes (tieredImagenet-S), where 10% of original meta-training classes, i.e, 35 classes, are used. The results show that MLTI outperforms other methods, further verifying its effectiveness.
>
> **Table R-1**: Results (Accuracies with 95% confidence) on tieredImageNet.
>
> |          | Strategies   |   tieredImageNet-S  |                   |
> |----------|--------------|:-----------------:|:-----------------:|
> |          |              |       1-shot      |       5-shot      |
> | MAML     | Vanilla      | 42.20 $\pm$ 0.84% | 58.23 $\pm$ 0.77% |
> |          | Meta-Reg     | 42.87 $\pm$ 0.86% | 59.16 $\pm$ 0.79% |
> |          | TAML         | 42.86 $\pm$ 0.84% | 59.33 $\pm$ 0.76% |
> |          | Meta-Dropout | 41.94 $\pm$ 0.82% | 58.37 $\pm$ 0.77% |
> |          | MetaMix      | 43.40 $\pm$ 0.85% | 61.92 $\pm$ 0.80% |
> |          | Meta-Maxup   | 43.69 $\pm$ 0.88% | 60.00 $\pm$ 0.82% |
> |          | **MLTI (ours)**  | **44.32 $\pm$ 0.82%** | **62.22 $\pm$ 0.79%** |
> | ProtoNet | Vanilla      | 43.35 $\pm$ 0.82% | 59.98 $\pm$ 0.77% |
> |          | MetaMix      | 44.14 $\pm$ 0.83% | 60.97 $\pm$ 0.81% |
> |          | Meta-Maxup   | 44.40 $\pm$ 0.83% | 61.79 $\pm$ 0.78% |
> |          | **MLTI (ours)**  | **45.47 $\pm$ 0.86%** | **62.35 $\pm$ 0.80%** |
>
> ---------
>
> **Q2**: Experiments on other domains
>
> **A2**: The experiments in the original submission already span multiple domains including computer vision, healthcare, and biology. Here, NCI and Metabolism use 1,024-dimensional Morgan fingerprint features as input and aim to predict the property of molecules. Tabular Murris uses gene expressions as input for each cell and aims to classify the types of cells. Detailed descriptions of these datasets are discussed in Appendix D.1 and E.1 in the original submission.
>
> We have now conducted additional experiments on few-shot text classification tasks (non-label-sharing setting). Following [Bao et al. 2020], we use Huffpost dataset to evaluate the performance and the results are listed in Table R-2 and Appendix E.8 (Table 14). We adopt ALBERT [Lan et al. 2019] as the pre-trained encoder and a classifier with two fully connected layers as the base model. Detailed dataset descriptions and hyperparameter settings are discussed in Appendix E.8.
>
> **Table R-2**: Results (Accuracies with 95% confidence intervals) on Huffpost.
>
> |          |              |      Huffpost     |                   |
> |----------|--------------|:-----------------:|:-----------------:|
> |          |              |       1-shot      |       5-shot      |
> | MAML     | Vanilla      | 39.51 $\pm$ 1.07% | 50.68 $\pm$ 0.90% |
> |          | Meta-Reg     | 40.32 $\pm$ 1.05% | 50.96 $\pm$ 0.98% |
> |          | TAML         | 40.03 $\pm$ 1.00% | 50.89 $\pm$ 0.88% |
> |          | Meta-Dropout | 39.89 $\pm$ 0.98% | 51.03 $\pm$ 0.91% |
> |          | MetaMix      | 40.64 $\pm$ 1.02% | 51.65 $\pm$ 0.92% |
> |          | Meta-Maxup   | 40.39 $\pm$ 1.01% | 51.80 $\pm$ 0.91% |
> |          | **MLTI (ours)**  | **41.06 $\pm$ 1.04%** | **52.53 $\pm$ 0.90%** |
> | ProtoNet | Vanilla      | 41.85 $\pm$ 1.01% | 58.98 $\pm$ 0.92% |
> |          | MetaMix      | 42.27 $\pm$ 0.98% | 60.43 $\pm$ 0.90% |
> |          | Meta-Maxup   | 42.39 $\pm$ 1.01% | 60.27 $\pm$ 0.88% |
> |          | **MLTI (ours)**  | **42.74 $\pm$ 0.96%** | **61.09 $\pm$ 0.91%** |
>
> According to the results, we find that our model also outperforms other strategies in Huffpost, demonstrating its effectiveness.
>
> ---------
>
> **Q3**: Theoretical Analysis
>
> **A3**: We have revised our paper in light of your comments by adding more explanations in the theoretical analysis section. Specifically, we’ve now stated more explicitly that the loss function induced by MLTI is approximately the standard loss function plus a positive term. We explained that such a positive term is a quadratic function on the parameter phi_i’s, and therefore can be interpreted as a regularization term. We further added more explanations around Theorem 1, and pointed out that the purpose of Theorem is to show how this regularization term improves the generalization error bound. Please kindly let us know if you have any further questions or concerns.
>
> ---------
>
> **References**
>
> [Bao et al. 2020] Bao, Yujia, et al. "Few-shot text classification with distributional signatures." ICLR 2020.
>
> [Lan et al. 2019] Lan, Zhenzhong, et al. "Albert: A lite bert for self-supervised learning of language representations." arXiv preprint arXiv:1909.11942 (2019).

---

### Public Comment · ~John_Doe2 · 2021-11-14
**Questions to reproduce results**

Dear authors,
We have been trying to reproduce the experiments in the paper. In the process, we have come across the following questions:
- How were the hyperparameters selected for datasets where no (meta-)validation set is given? No validation data were specified for Pose, ISIC, DermNet-S, NCI, and TDC. For miniImagenet-S, it is unclear whether the miniImangenet validation set is used. There is a mention of cross-validation in Appendix C.1, but it is unclear how and on what data this was carried out.
- Metrics: How many episodes were used to calculate the reported metrics and confidence intervals, and were the intervals calculated across repeated experiments or across tasks? How large were the query sets during evaluation?
- NCI: There exist two versions of the dataset: A "balanced" and a "full" one. From the reported accuracies, it appears that a balanced dataset was used. Is this the balancing from the "NCI balanced" dataset or was the full dataset balanced manually?
- TDC: Appendix C.1 says "We balance each subdataset by randomly selecting at most 1000 data samples" -- in this context, we are unsure about the specific meaning of "balance": equal number of True vs. False labels, or equal number of samples across tasks?
- DermNet:
  - Prabhu et al. (2018) note that they performed perceptual image hashing to remove duplicate images. Was this also done here, and if so, with which hashing algorithm?
  - Which model architecture was used for this dataset? The Resnet50v2 as used by Prabhu et al. (2018), or a different one?
- RainbowMNIST: According to Finn et al. (2019), each task contains 900 samples. Were these samples drawn only from the 60K MNIST train images or also from the 10K test images?

We would be grateful for your clarifications regarding these questions. Many thanks in advance.

---

> ### Author Response · Authors · 2021-11-16
> **Response to Questions about Reproducibility**
>
> Thanks for your interest and comments. We here reply to your questions one-by-one, and we also revised our paper to include more implementation details according to your comments.
>
> - Hyperparameters: Yes, we apply cross-validation to select hyperparameters. Following the traditional process of cross-validation, we randomly select some classes/samples for validation.
> - Metrics: Following the traditional setting in MAML [Finn et al. 2017], the intervals are calculated across all tasks. The maximum iterations and the query set size during evaluation have been reported in Table 5 and 7 in the Appendix of the original submission. The query set size in testing is the same as training. The optimal iteration to calculate the reported metrics is also determined by cross-validation.
> - NCI: The balanced dataset is generated from “NCI balanced”.
> - TDC: As discussed in Appendix E.1, we balance the data by randomly selecting at most 500 positive and 500 negative samples. If one subdataset has less than 500 positive/negative samples, we will use all positive/negative samples.
> - DermNet:
>     - We do not apply hash algorithms to remove images. Instead, we remove duplicate classes.
>     - The base model used in DermNet is the same as miniImagenet-S and ISIC, where a convolutional neural network with four convolutional blocks is used. We have put these details in Appendix E.1.
> - RainbowMNIST: we combine the training and test set of original MNIST data and randomly select 5,600 samples for each class. Each task has 1,000 samples in our experiments.
>
> **Reference**
>
> [Finn et al. 2017] Finn, Chelsea, Pieter Abbeel, and Sergey Levine. "Model-agnostic meta-learning for fast adaptation of deep networks." In International Conference on Machine Learning, pp. 1126-1135. PMLR, 2017.

---

### Author Response · Authors · 2021-11-16
**Summary of Paper Revision**

We appreciate all the reviewers for their valuable and constructive comments. According to these comments, we have improved the paper and summarized the main changes as follows:

1. Included the comparison between MLTI and other strategies on full-size miniImagenet and DermNet (Appendix E.2)
2. Added the results of Huffpost -- a few-shot text classification dataset, and tieredImagenet (Appendix E.8)
3. Provided base model analysis and interpolation layer analysis (Appendix E.5)
4. Revised the dataset descriptions and hyperparameter settings to include more details (Appendix D.1 and E.1)
5. Added more discussion on previous works in Appendix C, and more explanation of the theoretical analysis in Section 4.
6. Since more results and discussions are included in the Appendix, some Appendix indexes have been changed accordingly.

---

### Decision · Program_Chairs · 2022-01-20

**Decision:**

Accept (Oral)

**Comment:**

Current meta-learning algorithms suffer from the requirement of a large number of tasks in the meta-training phase, which may not be accessible in real-world environment. This paper addresses this bottleneck, introducing a cross-task interpolation in addition to the existing intra-task interpolation. The main idea is very simple, which can be viewed as an incremental adding-up to existing augmentation methods. However, the method is well supported by nice theoretical results which highlight the relation between task interpolation and the Rademacher complexity. In fact, this is not a trivial extension of existing work.  Authors did a good job in the rebuttal phase, resolving most of concerns raised by reviewers, leading that two of reviewers raised their score. All reviewers agree to champion this paper. Congratulations on a nice work.